# Limited physical protection leads to high organic carbon reactivity in anoxic Baltic Sea sediments

Silvia Placitu[1], Sebastiaan J. van de Velde[2,3,4], Astrid Hylén[4,5], Mats Eriksson[6], Per O.J. Hall[5], Steeve Bonneville[1]

[1] Department of Geosciences, Environment and Society, Université libre de Bruxelles, Brussels, Belgium

[2] Department of Marine Science, University of Otago, Dunedin, New Zealand

[3] National Institute of Water and Atmospheric Research, Wellington, New Zealand

[4] Department of Biology, University of Antwerp, Wilrijk, Belgium

[5] Department of Marine Sciences, University of Gothenburg, Gothenburg, Sweden

[6] Department of Health, Medicine and Caring Sciences, Linköping University, Linköping, Sweden

*Correspondence to*: Silvia Placitu (silviaplacitu@ulb.be)

**Abstract**

Marine sediments bury ~160 Tg organic carbon (OC) $yr^{-1}$ globally, with ~90% of the burial occurring in continental margin sediments. It is generally believed that OC is buried more efficiently in sediments underlying anoxic bottom waters. However, recent studies revealed that sediments in the central Baltic Sea exhibit very high OC mineralization rates and consequently low OC burial efficiencies (~5-10%), despite being overlaid by long-term anoxic bottom waters. Here, we investigate factors contributing to this unexpectedly high OC mineralization rates in the Western Gotland Basin (WGB), a sub-basin of the central Baltic Sea. We sampled five sites along a transect in the WGB, including two where organic carbon-iron (OC-Fe) associations were quantified. Sulphate reduction rate measurements indicated that OC reactivity ($k$) was much higher than expected for anoxic sediments. High OC loadings (i.e., OC concentrations normalized to sediment specific surface area) and low OC-Fe associations showed that physical protection of OC is limited. Overall, these results suggest that the WGB sediments receive large amounts of bulk OC relative to the supply of mineral particles, far exceeding the potential for OC physical protection. As a result, a large fraction of OC is free from associations with mineral surfaces, thus the OC reactivity is high, despite anoxic bottom waters. Overall, our results demonstrate that anoxia does not always lead to lower OC mineralization rates and increased burial efficiencies in sediments.

## 1 Introduction

OC burial within marine sediments modulates atmospheric oxygen ($O_2$) and carbon dioxide ($CO_2$) concentrations over geological timescales (Berner, 1982). Marine sediments are the largest active OC sink with ~160 Tg OC buried annually on a global scale (Berner, 1982; Burdige, 2007; Hedges and Keil, 1995; LaRowe et al., 2020). OC burial efficiency, which corresponds to the percent of the OC deposition flux to the sediments that is eventually preserved (and therefore, not mineralized) is low globally, with on average ~10% of the OC settling at the sediment-water interface being buried (Burdige, 2007). However, this number can reach up to 80% in continental shelf and coastal sediment depocenters (Blair and Aller, 2012; Canfield, 1994; Henrichs, 1992; Ingall and Cappellen, 1990; Ståhl et al., 2004). Multiple factors influence OC burial and mineralization rates (Arndt et al., 2013; LaRowe et al., 2020) and specifically, the availability of $O_2$ in the benthic environment has been suggested to play a pivotal role in regulating the sediment biogeochemistry and the structure of the faunal community, which in turn affects the efficiency and pathways of OC remineralization and burial (Kristensen et al., 2012; Middelburg and Levin, 2009). However, the extent to which $O_2$ influences the OC burial efficiency remains a topic of debate (Canfield, 1994; Middelburg and Levin, 2009; van de Velde et al., 2023). It is often reported that sediments deposited under anoxic bottom waters tend to exhibit apparent higher OC preservation.

This pattern is generally attributed not to the effective preservation of organic carbon in anoxic environments, but rather as the result of $O_2$ strongly enhancing OC mineralization when present. Several processes explain this : (i) aerobic respiration yields more free energy than anaerobic pathways, leading to faster degradation (LaRowe and Van Cappellen, 2011), (ii) certain complex organic molecules, such as lignin, can be more efficiently degraded via oxidative enzymes available only under aerobic conditions (Megonigal et al., 2003; Burdige, 2007; Canfield, 1994), (iii) aerobic organisms are capable of degrading OM completely to $CO_2$ (via the Krebs cycle) while mineralization under anoxic condition proceeds through a multi-step anaerobic food chains involving syntrophic interactions among specialized consortia which makes the degradation slower and less efficient (Arndt et al., 2013) ; (iv) oxygenated sediments support macrofauna that can directly or indirectly stimulate mineralization through bioturbation and bioirrigation and prevent accumulation of reduced toxic products such as $H_2S$ (Kristensen et al., 1992; Aller and Aller, 1998; Papaspyrou et al., 2007; van de Velde et al., 2020).

However, determining a clear relationship between burial efficiency and $O_2$ exposure is challenging, as the OC mineralization can be hindered by physical protection mechanisms, such as adsorption on – or coprecipitation with – organic and inorganic matrices, forming 'geomacromolecules' highly resistant to enzymatic hydrolysis and bacterial mineralization (Jørgensen, 2006). Examples of such processes are the formation of authigenic minerals, adsorption to mineral surfaces, encapsulation by refractory macromolecules or complexation with metals (Burdige, 2007; Henrichs, 1992; Lalonde et al., 2012). OC loading, defined as the ratio between OC content and specific surface area (SSA) of sediment, reflects the amount of OC associated with the total mineral surface available – including clays, Al, Mn, Fe oxides and other mineral phases – and is commonly used as a proxy to understand the OC fate in sediments. Because of their high SSA and positive surface charge, which confer enhanced adsorption capacity, reactive iron oxides ($Fe_R$) have been recognized for their significant role in sorbing OC in soils

and sediments (Barber et al., 2014; Faust et al., 2021; Lalonde et al., 2012; Mehra and Jackson, 1958; Placitu et al., 2024; Yao et al., 2023).

Recently, studies in the Baltic Sea suggested that the magnitude of the positive effect of $O_2$ level on OC mineralization rates has been overestimated (Nilsson et al., 2019; van de Velde et al., 2023). Extensive regions of the Baltic Sea suffer from prolonged periods of anoxia ($O_2 < 0.5\,\mu M$) due to eutrophication caused by intense land activities in the drainage area, strongly stratified waters induced by a permanent halocline around 60–80 m depth and limited water exchange with the North Sea, which lead to a relatively long water residence time of 25-35 years (Hall et al., 2017). Despite the absence of $O_2$ in bottom waters over a large area of the central Baltic Sea, carbon budget calculations indicate that ~22 Tg OC is mineralized and recycled back to the water column annually, while the long-term burial only involves $1.0 \pm 0.3$ Tg OC (Nilsson et al., 2019). This suggests that sedimentary OC in the Baltic Sea is more reactive than the environmental conditions would predict. van de Velde et al. (2023), combining *in situ* benthic chamber lander and core-based measurements, showed that the OC mineralization rates in anoxic sediments in the central Baltic Sea are much higher, and OC burial efficiencies much lower, than previously reported for sediments underlying anoxic bottom waters. Here, we evaluate how physical protection influences the mineralization and burial of OC by investigating five sites along a transect across the Western Gotland Basin (WGB) in the central Baltic Sea. We report OC decay rate constants ($k$) and OC loadings from five stations and quantify the OC-Fe associations (the so-called 'rusty carbon sink') at two stations representing contrasting redox conditions: one deep, persistently anoxic site and one comparatively shallow, hypoxic station.

## 2 Materials and Methods

### 2.1 Study area and sampling

The Baltic Sea is the largest brackish water body in the world (Björck, 1995), spanning from latitude 53°N to 66°N and longitude 10°E to 30°E, covering an area of approximately 370 000 km$^2$. Enclosed by the landmasses of Eastern and Central Europe and Scandinavia (Andrén et al., 2000), it connects to the North Sea via the Danish-Swedish straits, Öresund and Store Bælt. The vast drainage basin extends nearly 2 million km$^2$ and is home to approximately ~85 million people (Nilsson et al., 2021 and references therein).

During a cruise in August 2021 aboard the R/V Skagerak, five sites (named WGB1 to WGB5) were sampled along a transect in the Western Gotland Basin, WGB (Figure 1, Table 1). Bottom-water temperature, salinity and $O_2$ were recorded through a CTD instrument (SBE 911, Sea-Bird Scientific) equipped with a high-accuracy $O_2$ sensor (SBE 43, Sea-Bird Scientific). Sediment cores were retrieved using a GEMAX gravity corer (9 cm inner diameter). Two cores from each station were sectioned at 0.5 cm resolution from 0 to 2 cm depth, at 1 cm resolution between 2 and 6 cm depth, and in 2 cm slices from 6 to 20 cm depth in ambient air and samples were frozen until determination of the specific surface area (SSA) of the sedimentary particles, sedimentary organic carbon (SOC), nitrogen (N), and $\delta^{13}C$ signature. At stations WGB1 and WGB2, two extra sediment cores were collected for the determination of OC-Fe associations. These cores were processed within a glovebag

(Captair Pyramid, Erlab, France) under $N_2$ atmosphere immediately after collection. Slicing was done at 1 cm intervals to a depth of 15 cm. Sediment sections were then transferred to 50 mL Falcon tubes (VWR, USA) and frozen until analysis.

95    Determination of porewater sulphate ($SO_4^{2-}$), sulphate reduction rates (SRR), porosity, and $^{210}Pb$ dating are detailed in van de Velde et al. (2023). Briefly, two cores at each station were sectioned under $N_2$ atmosphere in a portable glove bag (Captair Pyramid, Erlab, France) at the same resolution as the SOC cores and porewater was collected using Rhizon samplers (pore size ~0.15 µm; Rhizosphere Research Products, The Netherlands). Porewater samples for $SO_4^{2-}$ analysis were stabilized using ZnAc (2.25 ml of a 10% ZnAc solution per 0.25 ml sample) and stored at 4°C. Two subcores (2.5 cm inner diameter, 20 cm

100   length) were collected from the GEMAX corer at each station for SRR measurements, using the $^{35}S$ radiotracer method (Jørgensen, 1978) immediately on retrieval, and the tracer addition and incubation started within 15 minutes of core collection. Duplicate cores were collected from two different GEMAX casts. One core per station was collected for $^{210}Pb$ dating at Linköping University, Sweden.

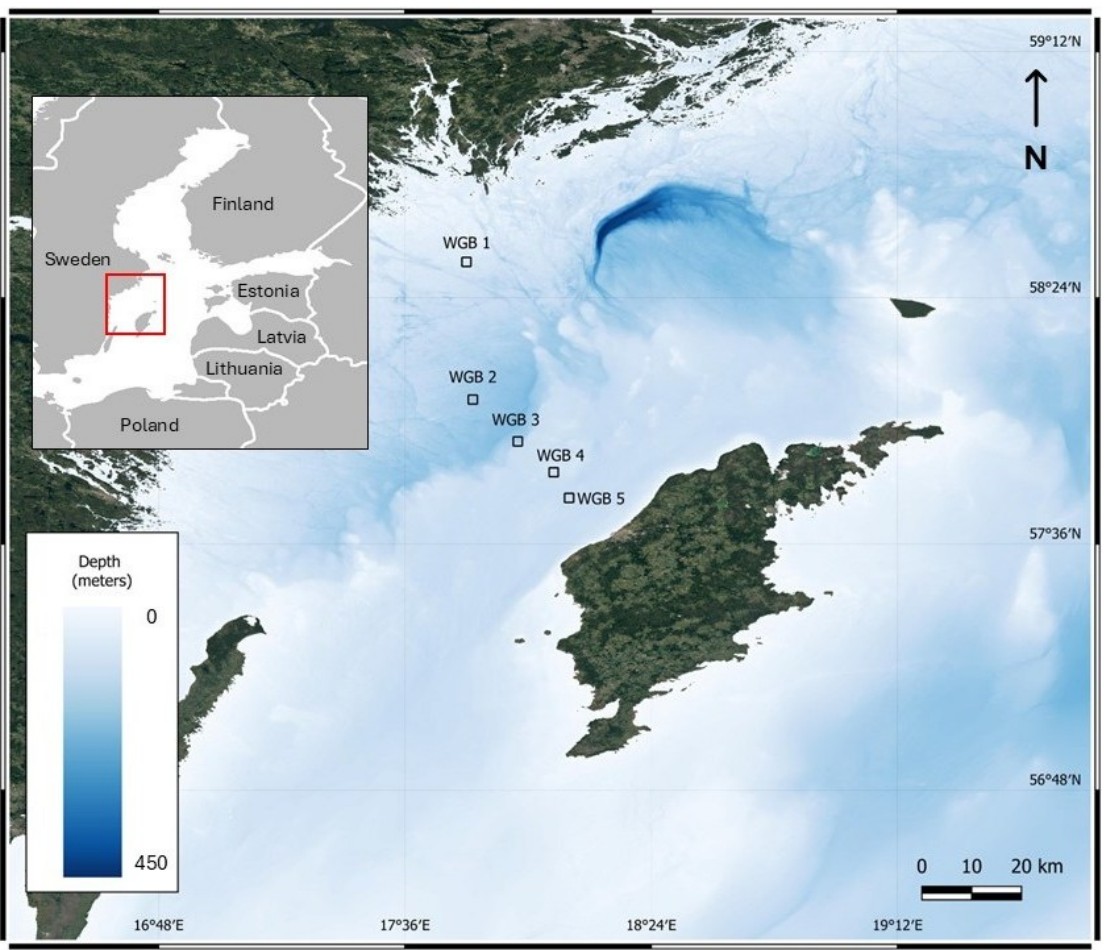

**Figure 1: Sampling locations in the Western Gotland Basin, Baltic Sea. The bathymetry is from https://emodnet.ec.europa.eu/en**

**Table 1: Sampling location and bottom water CTD data. Mass accumulation rates (MAR) are taken from van de Velde et al. (2023).**

| Station | Coord. | Water depth m | Sal. | Temp. °C | [$O_2$] BW μM | MAR g m$^{-2}$yr$^{-1}$ |
|---------|--------|---------------|------|----------|---------------|-------------------------|
| **WGB1** | N 58°31 E 17°48 | 75 | 10 | 5.7 | 0-20 | 108–285 |
| **WGB2** | N 58°04 E 17°49 | 170 | 11 | 6.3 | anoxic | 59 – 285 |
| **WGB3** | N 57°56 E 17°58 | 160 | 11 | 6.2 | anoxic | 84–226 |
| **WGB4** | N 57°50 E 18°05 | 100 | 11 | 6.2 | anoxic | 70–230 |
| **WGB5** | N 57°45 E 18°08 | 110 | 11 | 6.1 | anoxic | 60–180 |

## 2.2 SOC, N, δ$^{13}$C signature and OC loadings

Prior to determination of SOC, N, and isotopic signature, sediments were freeze dried and homogenized by gentle grinding in an agate mortar, which was cleaned with acetone after each sample. Typically, 15 to 30 mg of dry sediment were weighed in a silver capsule and decarbonated via fumigation overnight in a desiccator using 37% HCl fuming acid after adding a drop of MilliQ water to the sediment (Harris et al., 2001). The samples were then dried under an infrared lamp and transferred into tin capsules before being loaded into on an elemental analyser (Sercon Europa EA-GSL; Sercon Ltd., Crewe, UK) coupled to an isotope ratio mass spectrometer (IRMS; Sercon 20-22) at the ISOGOT facility (Department of Earth Sciences, University of Gothenburg). A certified reference material (high organic content sediment standard; Elemental Microanalysis, UK) was used as an internal standard throughout analysis, approximately every 10-12 samples. This standard has a certified isotope value of δ$^{13}$C -26.27 ‰. Drift was corrected relative to the standard and reproducibility was determined from replicate standard analyses. Blanks were run at the beginning of each analytical run to verify negligible contamination from previous analyses.) Linearity was assessed during annual service using reference gases ($CO_2$).

SOC and N contents are expressed as % of sediment dry weight, and the δ$^{13}$C isotopic signature are reported as ‰ deviations from Vienna Pee Dee Belemnite (VPDB) standard. The relative precision of the SOC measurements was 5% or better.

The SSA of sediments was determined for certain sediment depths (See Dataset in Supplementary Information) on an aliquot of the SOC samples, using the 5-point BET method (BET, $N_2$ adsorption isotherm at 77K) via ASAP 2020 surface area analyser (Micromeritics, US). Prior to the SSA analysis, freeze-dried sediment samples were heated at 350°C for 24 h to remove OC following the method described by Cui et al. (2022). The OC loadings (mg C $m^{-2}$) are calculated as the ratio between SOC content (mg C $g^{-1}$) and SSA ($m^2$ $g^{-1}$).

## 2.3 Extraction and calculation of OC-Fe

The OC-Fe associations were quantified in sediment cores from WGB1 (the shallowest station with hypoxic BW) and WGB2 (the deepest station with anoxic BW) which together bracket the environmental conditions present across the WGB, using the citrate-bicarbonate-dithionite (CBD) extraction method adapted from Mehra and Jackson (1958) described in Lalonde et al. (2012) . This extraction method targets all reactive Fe ($Fe_R$), while leaving clay minerals unaffected (Lalonde et al., 2012). In brief, 250 mg of freeze-dried sediment were weighed and placed into a 50 mL Falcon tube. Then, 15 mL of the extraction solution (0.27 M trisodium citrate and 0.11 M sodium bicarbonate) were added, and the tube was placed in a water bath at 80°C. Once the temperature was reached, 250 mg of sodium dithionite were introduced to the solution which was continuously agitated in the water bath to maintain a constant temperature. After 15 min, the extraction was stopped and the tube centrifuged (Eppendorf centrifuge 5804, Germany) at 3200 g for 10 min, after which the supernatant was filtered through a 0.22 μm PTFE filter and collected in 50 mL Falcon tubes. The sediment was subsequently rinsed with 10 mL of artificial seawater (0.235 M sodium chloride and 0.0245 M magnesium sulphate heptahydrate), centrifuged, and filtered. The rinsing step was repeated three times. The rinsing and the extraction solutions were combined and acidified with 100 μL trace-metal grade HCl (32%) to prevent Fe reoxidation.

Parallel to the CBD extraction, a control extraction was performed to discriminate the portion of OC not associated with $Fe_R$, but solubilized due to the high ionic strength of the solution (Fisher et al., 2020). In the control extraction, trisodium citrate was substituted with 1.6 M of sodium chloride, and the sodium dithionite was substituted with 220 mg of sodium chloride to maintain the same ionic strength as the CBD solution. The extraction protocol mirrored the CBD extraction, with all steps described in (Table S1 in Supplementary Information). The Fe concentrations in the supernatant solutions were determined by ICP-OES (Varian, VISTA-MPX CCD, US) with an analytical precision of 5%. Sediment leftovers from the extractions were frozen at -20°C and then freeze dried prior to SOC analyses as described above.

The percentage of bulk OC associated with $Fe_R$ (%OC-Fe) is calculated as the difference between the %OC left after the control extraction (%$OC_{Control}$) and the %OC residual post CBD extraction (%$OC_{CBD}$), normalized to the total OC in the bulk sediment (%$OC_{Bulk}$) before extractions as follow:

$$\%OC - Fe = \frac{\%OC_{Control} - \%OC_{CBD}}{\%OC_{Bulk}} 100 \qquad (1)$$

## 2.4 OC reactivity $k$ and [210]Pb-estimated OC age

The OC reactivity, approximated by the first-order decay rate constant $k$ (yr$^{-1}$), is derived from the volumetric sulphate reduction rates (SRR). The rate of OC mineralization $R_{min}$ (nmol cm$^{-3}$ yr$^{-1}$) at depth x (cm) can be calculated as:

$$R_{min\ (x)} = k_{(x)}C_{(x)} \qquad (2)$$

where $C_{(x)}$ represents the OC concentration (nmol cm$^{-3}$) at sediment depth x. These sediments are anoxic, unaffected by bioturbation, and sulphate SO$_4^{2-}$ is not depleted, so one can assume that all OC mineralization results from sulphate reduction. Contributions from denitrification, DNRA, manganese and iron reduction are negligible; however, it is important to note that the reported rates may slightly underestimate total reactivity:

$$2CH_2O + SO_4^{2-} \rightarrow 2HCO_3^- + H_2S \qquad (3)$$

The value of $R_{min(x)}$ can thus be approximated by the SRR, accounting for the 2 C for 1 S stoichiometry during sulphate reduction:

$$k_{(x)} = \frac{2SRR_{(x)}}{C_{(x)}} \qquad (4)$$

Since these WGB sediments are unbioturbated, the SOC age (yr) can be approximated from the [210]Pb dating of each sediment depth. However, as the [210]Pb dating quantifies the time since deposition at the sediment surface, it underestimates the actual age of the SOC (i.e., time since formation).

## 3 Results and Discussion

### 3.1 Organic carbon in Western Gotland Basin (WGB) sediments is highly reactive

The sedimentary organic carbon (SOC) profiles (Figure 2, A-E) generally decrease with depth at all stations, but with distinct site-specific trends. At WGB2 and WGB3, the deepest stations (Table 1), surface SOC contents reach up to 20 wt%, while WGB1 shows lower surface values and a more gradual decline at depth. At WGB4, an increase of SOC at ~5 cm depth may suggest a potential, localized input of refractory OC (possibly of terrestrial origin). Elevated SOC concentrations in the Baltic Sea generally stem from cyanobacteria, diatoms, and other phytoplankton blooms induced by nutrients inputs, as indicated by δ[13]C signatures and C/N ratios of WGB1 and WGB2 sediments (Figure S1 in Supplementary information), reflecting autochthonous freshwater and marine inputs.

At WGB1, progressive δ[13]C depletion with depth and increasing C/N indicate selective microbial degradation of labile, N-rich compounds (Figure S1 in Supplementary information). At WGB2, δ[13]C values fluctuate irregularly with depth (Figure S1 in Supplementary information). Such variability could, in principle, reflect changing rates of microbial SOC degradation, as

isotopic fractionation during microbial metabolism generally enriches the residual organic matter in $^{13}C$. However, the $\delta^{13}C$ and SRR profiles in WGB2 show no consistent covariation, indicating that variations in microbial activity are unlikely to be the primary driver of the $\delta^{13}C$ pattern. Instead, the variations of $\delta^{13}C$, C/N and SRR most likely reflect heterogeneity in the deposited material: certain layers may contain more reactive OC that transiently enhances microbial activity, whereas others consist of more degraded OC that experienced substantial mineralization during lateral transport. This interpretation aligns with the WGB sedimentation regime, where lateral transport involves repeated resuspension from shallow, erosive regions (e.g., WGB1) and redeposition in deeper, accumulating depocenters (e.g., WGB2; Nilsson et al., 2021). In addition to this transport, since the mid 1900's, enhanced nutrient loading in the central Baltic Sea has increased OC inputs, while damming of major rivers reduced sediment delivery and limited dilution of autochthonous OC, contributing to the heterogeneity of SOC profiles in WGB (Emeis et al., 2000; Leipe et al., 2011). The SRR depth profiles (Figure 2, F–J) show generally elevated activity in the surficial sediments, consistent with the distribution of SOC. Stations WGB2 and WGB3 exhibit the highest SRR values, but with contrasting depth trends: WGB3 displays markedly higher surface rates that decline steeply within the upper 5 cm, whereas WGB2 shows more moderate but relatively uniform SRR throughout the upper layers. This difference, despite broadly similar SOC depth profiles, suggests that SRR is influenced not only by SOC content but also by other factors such as organic matter quality and degradability, microbial community and porewater geochemistry.

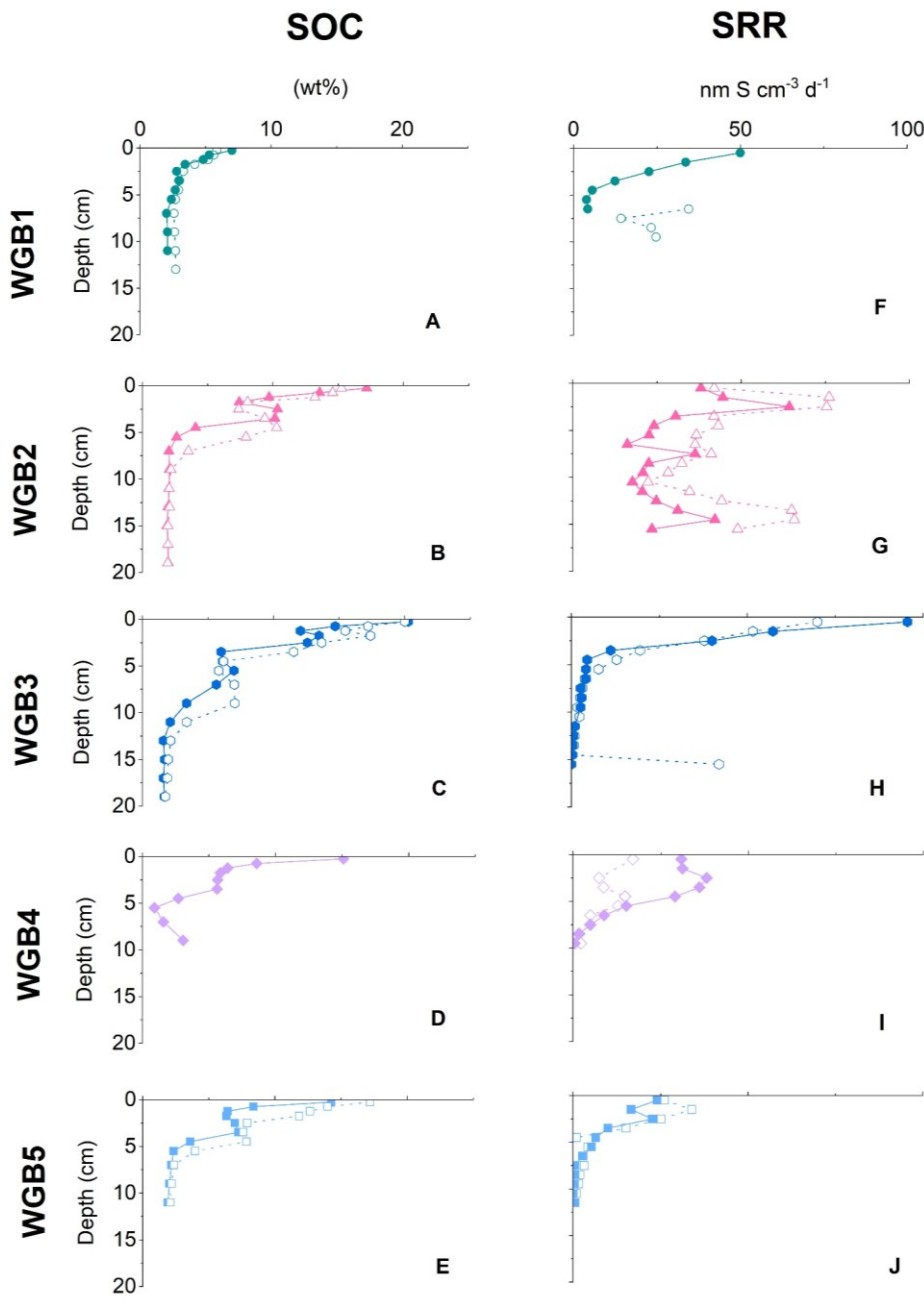

**Figure 2: Sedimentary organic carbon SOC (A-E), and sulphate reduction rates (SRR) profiles (F-J) at all locations. Full and empty symbols are duplicate cores.**

Generally, bulk OC reactivity approximated by the first-order decay constant $k$ (yr$^{-1}$) decreases with $^{210}$Pb-estimated OC age as the more reactive compounds are mineralized first, leaving behind the less reactive fractions (Middelburg et al., 1997). The $k$ values we found in the WGB sediments range from $10^{-1}$ to $10^{-3}$ yr$^{-1}$, consistent with previously reported values for sediments, that range from $10^{0}$ to $10^{-6}$ yr$^{-1}$ (Katsev and Crowe, 2015; Middelburg, 1989). These high OC reactivity values corroborate the very low OC burial efficiencies reported for the WGB (van de Velde et al., 2023). Katsev and Crowe (2015) demonstrated that

the relationship between the OC reactivity and the OC age differs between OC exposed to $O_2$ and OC mineralized under anoxic conditions (Figure 3), consistent with evidence that OC mineralization efficiency increases with $O_2$ exposure time (Hartnett et al., 1998; Hulthe et al., 1998). Because large parts of the WGB are long-term anoxic, the $k$ values would be expected to follow the reactivity-age relationship derived for anoxic sediments. Intriguingly, however, most of the $k$ values fall closer to the oxic reactivity-age trend line (Figure 3). Root Mean Squared Error (RMSE) for the anoxic trendline and the $k$ vs. $^{210}$Pb-estimated

OC age dataset is 0.68, while the RMSE for the oxic trendline was significantly lower at 0.45. This indicates that, overall, the sediment reactivity aligns more closely with the oxic trendline. Note, however, that the OC age estimated by the $^{210}$Pb method may be underestimated (see section 2.4) which would shift our reactivity-age dataset further to the right toward the oxic trendline.

       Importantly, the OC reactivity derived for oxic conditions in the compilation of Katsev and Crowe (2015) predominantly

consisted of water column data, whereas their anoxic conditions were taken from sedimentary environments. A large fraction of the particulate organic matter is fragmented into smaller fractions during the sinking to the benthic environment and is exposed to microbes, zooplankton, and nekton that consume the more reactive fractions. Consequently, the remaining particulate organic matter reaching the sediment is comparatively less reactive than its water column counterpart and it is often found sorbed to mineral surfaces or within aggregates with the sediment particles (Arndt et al., 2013; Burdige, 2007). The high

OC reactivity estimated in our study, despite anoxic conditions, suggests that the pattern observed by Katsev and Crowe (2015) could be driven by the interaction between OC and sedimentary particles, rather than a direct $O_2$ effect. Indeed, the high OC reactivity in the WGB sediments could be explained by (i) a lack of physical protection through adsorption to sediment particles, and (ii) a lack of protection by interactions with Fe minerals. We explore these factors below.

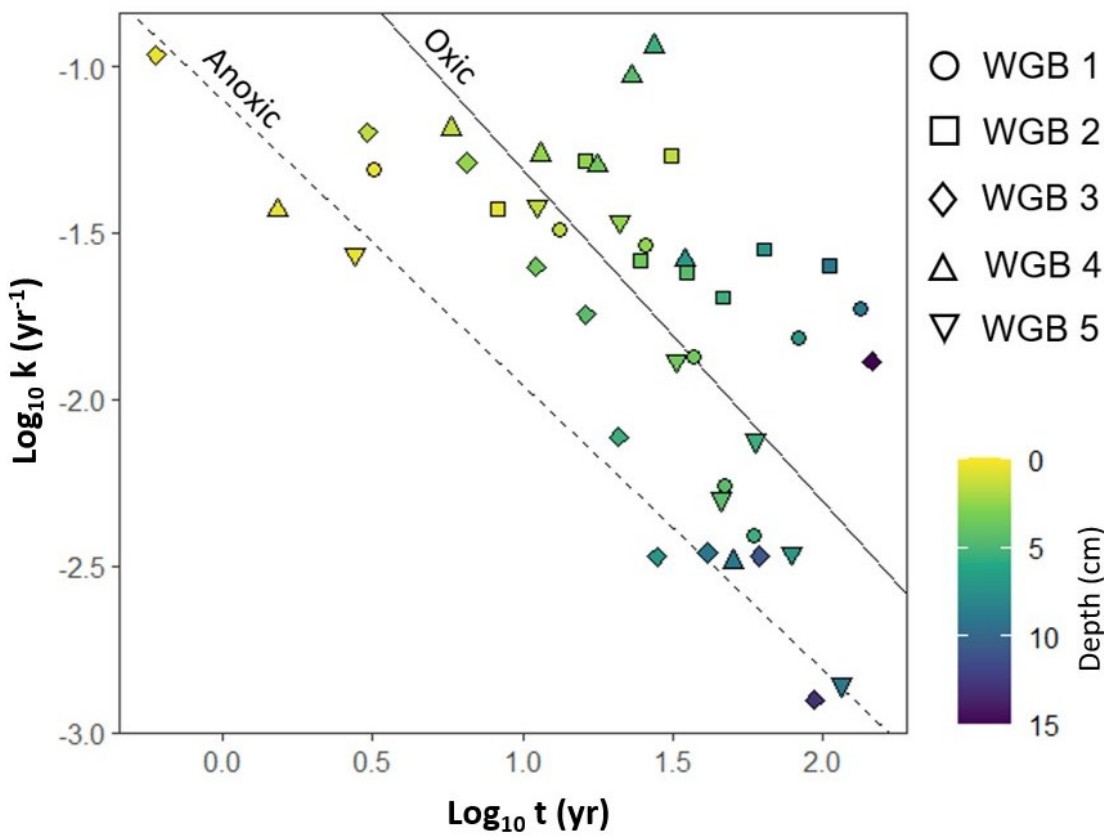

**Figure 3: OC reactivity (*k*) versus $^{210}$Pb estimated sediment age (t) in logarithmic scales. Data are represented per station using different symbols; individual points represent different sediment depths at a given station. The two regression lines are from Katsev & Crowe 2015, where the dashed line is for anoxic conditions (log(k) = -0.857log(t) -1.1), and the solid line is for oxic conditions (log(k) = -0.977log(t)-0.312).**


## 3.2 OC loading and physical protection

Because of the often observed positive correlation between SOC content and sediment specific surface area (SSA), SOC is assumed to be primarily associated with particle surfaces, which is consistent with the stabilization of OC through sorption to mineral surfaces (Goni et al., 2008; Keil et al., 1994). While this mechanism has been well documented in terrestrial settings

(Blair and Aller, 2012; Goni et al., 2008), where sorption provides protection to inherently more refractory compounds such as lignin-rich material, it has also been demonstrated in marine sediments (Keil et al., 1994; Li et al., 2017; Mayer, 1994b). In latter case, sorption on mineral surfaces may act to stabilize more labile, phytoplankton-derived OC. Such physical protection presumably promotes OC burial, making OC loading an important proxy for the OC fate (albeit all OC included in the loading calculations is not necessarily adsorbed to mineral surfaces (Li et al., 2017; Mayer, 1994b). Local environmental conditions

determine the OC loading (Bianchi et al., 2018; Blair and Aller, 2012). Low OC loadings (<0.4 mg OC m$^{-2}$) are indicative of frequently resuspended sediments that possibly undergo disaggregation and are exposed to O$_2$ for prolonged periods, leading to efficient OC mineralization. OC loadings between 0.4 and 1 mg OC m$^{-2}$ are typical for river-suspended sediments and those found downcore in shelf sediments respectively. High OC loadings (> 1 mg OC m$^{-2}$) are commonly observed in sediments from high-productivity regions such as upwelling zones or areas experiencing eutrophication, which have a high OC delivery
relative to the detrital sediment input.

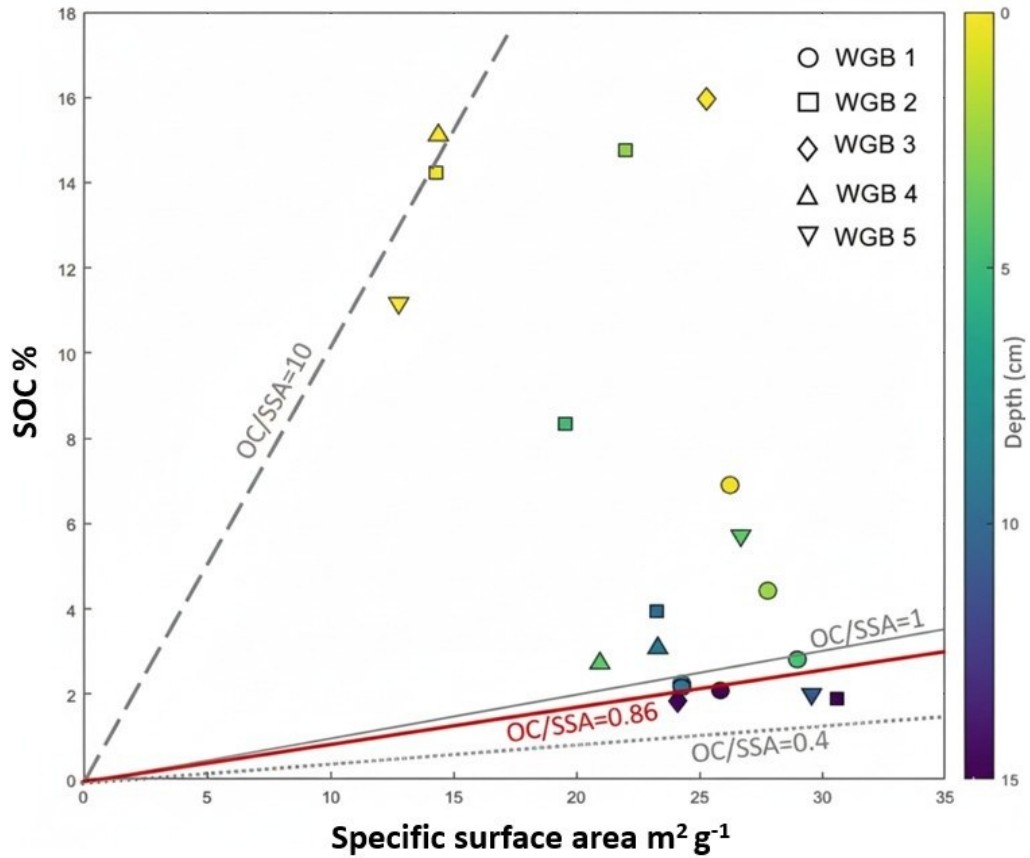

**Figure 4: Sedimentary organic carbon (SOC) content versus specific surface area (SSA). Stations are represented by symbols with filling colours indicating depth within profiles. Lines correspond to specific OC loadings (OC/SSA): 0.4, 1 and 10 mg OC m$^{-2}$. The bold red line corresponds to the monolayer equivalent adsorption (0.86 mg OC m$^{-2}$).**

The very high OC loadings observed in the WGB surface sediments most likely reflect a combination of intense autochthonous OC inputs, stimulated by eutrophication, together with a limited role of mineral surfaces in constraining OC accumulation.
OC loadings range from approximatively 2 mg OC m$^{-2}$ in the hypoxic WGB1 to over 6 mg OC m$^{-2}$ at other stations, peaking at around 10 mg C m$^{-2}$ in WGB4 (Figure 4). These values are comparable to those reported in upwelling regions or other settings with high productivity such as the Peruvian slope or Black Sea surface sediments (Mayer, 1994a). Enhanced nutrient

loading and primary production in the Baltic Sea during the 20th century (Conley et al., 2009) likely contributed to these elevated OC:SSA values in surficial sediments, as increased deposition of autochthonous OC on particle surfaces was not supported by a proportional increase in mineral inputs. With depth, OC loading systematically declines across all stations, dropping below 1 mg C m$^{-2}$ beneath 10 cm and falling below the monolayer equivalent adsorption threshold of 0.86 mg C m$^{-2}$ at ~15 cm (Mayer, 1994b). This downcore decrease parallels the sharp decline in both SOC concentrations and SRR (Figure 2) indicating that high surficial OC loadings represent reactive organic pools that are efficiently degraded rather than being stabilized by mineral associations. Thus, the combined OC/SSA, SRR, and OC% patterns point to a system where microbial activity dominates over mineral protection in regulating OC preservation in WGB sediments.

While direct observations of the OC distribution on particle surfaces in the sediment are lacking in this study, it has been demonstrated that OC does not form uniform coatings or infillings on sediment particles; rather, it likely exists as discrete blebs on the mineral surface or as non-associated organic debris (Ransom et al., 1997). The latter were shown to disproportionately contribute to OC loading (Bianchi et al., 2018) and we hypothesize that they form a large portion of the OC present in our profiles. High SOC due to eutrophication, together with intense mineralization and limited mineral surface protection (likely saturated) explains both the exceptionally high OC loadings at the surface and their rapid decline with depth. This underscores that OC preservation in WGB sediments is governed by both historical input dynamics and post-depositional degradation processes, rather than mineral protection alone.

## 3.3 The rusty carbon sink

The very high OC loadings observed in WGB sediments (section 3.2) indicate that mineral surfaces overall play a limited role in preserving OC from degradation. This interpretation is reinforced by the very low fraction of OC bound to Fe$_R$ in the two stations tested (Figure 5), compared to a global average for marine sediments of 15-20% (Lalonde et al., 2012; Longman et al., 2022; Zhao et al., 2018). At WGB1, Fe$_R$ binds on average 1.25 % of total OC (ranging from 0 to ~7 %OC-Fe), with OC-Fe associations restricted to the top 4 cm. The maximum of 7% OC-Fe occurs near the surface and decreases with depth, consistent with reductive dissolution of Fe phases in OC-Fe associations. This strongly contrast with the observations in the Barents Sea where OC-Fe associations persist over millennia (Faust et al., 2021). WGB1 also shows the highest %OC-Fe values despite low concentrations of both SOC (average of $3.5 \pm 1.3$ %) and Fe$_R$ (average of 14.7 µmol g$^{-1}$, ranging from 3 to 76 µmol. g$^{-1}$). This suggests that the formation of OC-Fe associations is not solely controlled by Fe$_R$ or SOC availability (Faust et al., 2021; Longman et al., 2021; Peter and Sobek, 2018; Sirois et al., 2018). Similar observations and conclusions have been drawn from Swedish fjord sediments where low and variable amounts of %OC-Fe with no relation with Fe$_R$ nor SOC concentrations have been reported in Placitu et al., (2024).

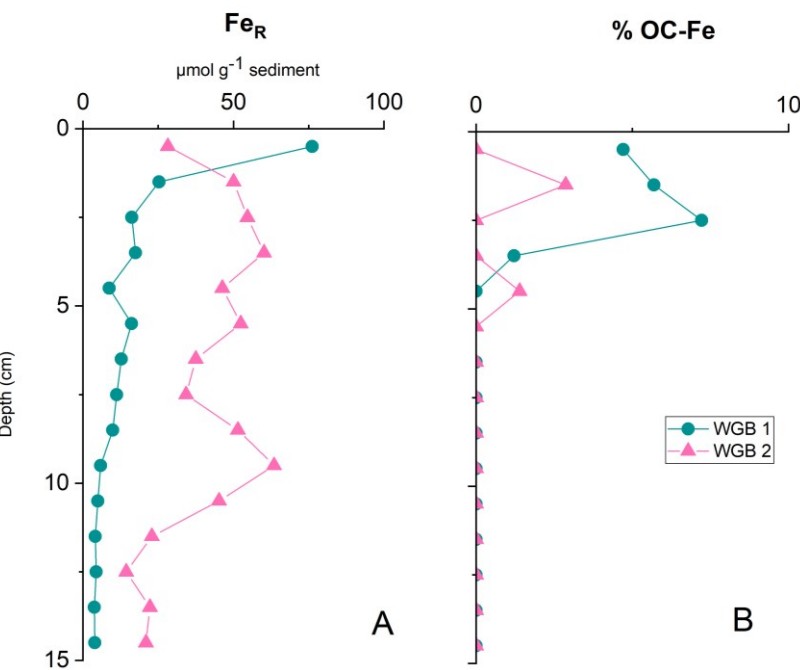

**Figure 5: Downcore profiles of reactive iron (Fe$_R$; A) and fraction of organic carbon associated to Fe$_R$ (%OC-Fe; B) at stations WGB1 and WGB2.**


At WGB2, despite higher SOC (6.4 ± 4.7%) and Fe$_R$ (average of 40.3 µmol. g$^{-1}$, ranging from 14 to 63 µmol. g$^{-1}$) than in WGB1, only 0.28 %OC-Fe (ranging from 0 to 2.8 %OC-Fe) is found. This is remarkable given that Fe$_R$ concentrations are similar to those reported for other O$_2$-depleted and eutrophic settings such as the Black Sea where Fe$_R$ binds ~15% of the OC (Lalonde et al., 2012). The very low %OC-Fe at WGB2 may reflect the effects of repeated physical reworking and sediment

redistribution prior to deposition in WGB2. Frequent resuspension and redeposition events can disrupt the OC-Fe associations by exposing particles to fluctuating redox conditions and mechanical disaggregation, as observed in mobile mud environments (Zhao et al., 2018). This process is consistent with the strong downcore variability in δ$^{13}$C and SRR at WGB2 and with earlier evidence of sediment shuttling and lateral transport in the area (Nilsson et al., 2021). The persistence of Fe$_R$ under anoxic and sulphidic conditions in WGB2 is intriguing. One potential explanation is the presence of Fe(II)-phosphate mineral phases such

as vivianite, which have been well-documented in Baltic Sea sediments and can be solubilized during the CBD extraction (e.g. Dijkstra et al., 2018; Egger et al., 2015; van Helmond et al., 2020; Kubeneck et al., 2021).

Although the downcore variability in our measurements indicates that depth trends should be interpreted with caution, a speculative interpretation of this downcore trend could be that the build-up of H$_2$S can reduce Fe$_R$ thus destabilizing the OC-Fe associations (Chen et al., 2014), which might ultimately precipitate as OC-Fe-sulphide associations. However, this so called

''black carbon sink'' has so far only been reported in laboratory studies (Ma et al., 2022; Picard et al., 2019). Differences in the %OC-Fe could also be due to mineralogical variations within the $Fe_R$ pool at depth, as the CBD extraction reduces and solubilizes a wide array of $Fe_R$, such as ferrihydrite, goethite, and lepidocrocite including Fe(II)-phosphates phases, all of which characterized by their small particle size, yet exhibiting large differences in SSA and surface reactivities towards OM moieties (Egger et al., 2015; Ghaisas et al., 2021). These characteristics determine the sorption efficiency, and therefore the

extent of the OC-Fe associations. For instance, OC associations with more crystalline $Fe_R$ minerals are believed to be less efficient in preserving organic matter from mineralization, creating mono or multi-layer sorption that are probably still accessible to microbial mineralization (Faust et al., 2021).

Taken together, these results indicate that Fe minerals contribute only marginally to the OC preservation in the WGB. While we cannot rule out that other mineral phases (e.g. clays, Al oxides) also provide sorption sites, the combination of very low

%OC-Fe and high OC loadings strongly suggest that physical protection is very limited in WGB. The low %OC-Fe found in WGB1 and WGB2 potentially reflects both the decoupled delivery of $Fe_R$ and OC and the absence of substantial terrestrial inputs of $Fe_R$ that preferentially bind terrestrial organic matter. In the WGB, $Fe_R$ is likely to enter the Baltic Sea via surface runoff and is redistributed via sediment resuspension as described in Nilsson et al., (2021), while most OC consists of autochthonous marine production (Figure S1 in Supplementary information) which likely limits the formation of OC-Fe

associations. This decoupling of OC and $Fe_R$ inputs in WGB could explain the limited amount of OC-Fe associations found across the WGB sediments.

## 4. Conclusion

Sediments in the WGB underly hypoxic and anoxic bottom waters, yet the reactivity of sedimentary OC is in line with previous

estimates from oxic settings. Detailed analyses at two representative sites (WGB1 and WGB2) show limited potential for physical protection of OC by adsorption on mineral surfaces and association with Fe oxides in these sediments, demonstrated by high OC loadings and low amounts of OC-Fe associations. Together with consistently high sulphate reduction rates across the basin, these results suggest that anoxic conditions in the WGB do not necessarily result in a slowdown of OC mineralization. Instead, the significant degradation of OC under anaerobic conditions is likely facilitated by the limited

physical protection of OC by mineral surface, which keeps a large portion of OC accessible to sulphate reducers. While our targeted dataset does not capture the full extent of spatial heterogeneity of the WGB, it nevertheless supports the view that mineral protection, rather than redox state alone, constitutes a critical control on the long-term burial of OC in marine sediments.

## Data availability

The datasets used in this study are available in the Zotero repository https://doi.org/10.5281/zenodo.17174549.

## Competing interests

The authors declare that they have no conflict of interest.

## Authors contribution

Conceptualization: SP, SJV, SB; Field sampling: SP, SJV, AH, POJH; Formal analysis: SP; Writing – original draft: SP;
Writing – review & editing: SP, SB, SJV, AH, ME, POJH; Funding acquisition: POJH, SB, SJV.

## Acknowledgements

The authors would like to thank the crew of the University of Gothenburg RV Skagerak for technical assistance, Elizabeth Robertson and Rebecca James for help at sea and with sample analysis, and Marie Carlsson (Linköping University) for her work with the $^{210}$Po analysis.

## Financial support

The research leading to this manuscript was supported by grants from the Swedish Research Council (grant no. 2015-03717 to POJH), the Swedish Agency for Marine and Water Management (grant no. 2535-20 to POJH), and the Belgian Federal Research Policy Office (grant no. RV/21/CANOE to SJV). AH was supported by a postdoctoral fellowship from the Research Foundation – Flanders (FWO; project nr 1241724N). SP and SB acknowledge the support of the Fonds National de la
Recherche Scientifique (FRS-FNRS PDR-T002019F).

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
