# Peer review of "Limited physical protection leads to high organic carbon reactivity in anoxic Baltic Sea sediments"

_EGUsphere, 2025_

## Referee Comment (RC1)

**EGUsphere-2025-3020 Manuscript Review: Limited physical protection leads to high organic carbon reactivity in anoxic Baltic Sea sediments**

**Summary**

To investigate the importance of mineral protection on OC reactivity in anoxic conditions, sediment cores were collected from the anoxic Baltic Sea. Their OC%, C/N ratio, $\delta^{13}C$, sulfate reduction rate, $^{210}Pb$ age, OC loading, and OC-Fe$_R$ were determined. The downcore increase in OC% and SRR suggested intensive sulfate reduction and OC remineralization, suggesting high OC reactivity despite the anoxic conditions. High OC loading, with downcore decrease in OC/SAA, suggests inadequate mineral surface area for OC stabilization. Low abundance of OC-Fe$_R$ indicated limited protection of OC by Fe$_R$ minerals. In summary, the current study demonstrates that when mineral protection of OC is nearly absent, oxygen depletion alone could not decrease OC mineralization rate.

**Major comments**

Overall, this manuscript is well-written. The data set presented in the current study is small and simple, yet it addresses an important research question in the chemical oceanography field. Therefore, I believe that the current manuscript contains a global significance that could be beneficial for further scientists who aim to investigate the drivers of OC preservation. Most of the methods and discussions are logical. I'd suggest this manuscript a minor revision, following my comments below.

**Specific comments**

**Line 23-24:** " Overall, these results suggest that the WGB sediments receive large amounts of OC relative to the supply of mineral particles, far exceeding the potential for OC physical protection"

This statement is true for OC/SAA; however, it might not be right for OC-FeR since the authors discussed that FeR availability is not the major factor controlling OC-FeR association (Line 267-273). Alternatively, I think the lack of OC-FeR in Baltic Sea is likely due to the separate input of FeR (from land) and OC (from freshwater/marine water column). See my comments in line 276-278.

**Line 74:** Add "," after "(Andrén et al., 2000)".

**Line 82:** In my opinion, "particulate organic carbon (POC)" is not an appropriate terminology here. Many literatures defined POC as suspended OC particles within water column, which has not deposited and has not been incorporated into sedimentary storage yet (see reviews in Bianchi (2007)). Another set of literature defined POC as a subset of bulk sedimentary OC that is made of fragmented plant litter with low density and large particle size (e.g., Cotrufo et al., 2019). To my understanding, the current study measured OC content in bulk sediment samples. Therefore, to avoid confusion, I'd suggest the authors to change the term "particulate organic carbon (POC)" to "sedimentary organic carbon (SOC)".

*Bianchi, T. S. (2007). Biogeochemistry of estuaries. Oxford University Press.*

*Cotrufo, M. F., Ranalli, M. G., Haddix, M. L., Six, J., & Lugato, E. (2019). Soil carbon storage informed by particulate and mineral-associated organic matter. Nature Geoscience, 12(12), 989-994.*

**Line 150:** What's about OC mineralization by reduction of $NO_3^-$ to $NH_4^+$, or reduction of $Fe^{3+}$ to $Fe^{2+}$?

**Line 164-165:** "...inputs, reflected in the $\delta^{13}C$ signature and C/N ratios of these sediments, highlighting the presence of 'fresh' material"

       $\delta^{13}C$ and C/N ratios are proxies of OC sources, not their freshness. I do agree with the authors that the principal sources of OC in WGB cores are freshwater and marine planktons. However, these proxies do not reflect whether these OC are made of 'fresh' plankton-derived OC, or 'degraded' plankton-derived OC. To determine whether these OC are fresh or degraded, the authors need additional proxies such as $\Delta^{14}C$ (age of OC) or biomarkers such as stanol/sterol ratios (higher stanol reflected more degraded OC).

**Line 182-183:** To my understanding, Katsev and Crowe (2015) calculated k-values from changes in concentration of C with depth and time, using the equation $k = -\frac{U}{c}\frac{dC}{dx}$. In contrast, the current study calculated k-values from SRR, through the measurement of $^{35}S$ radiotracer during laboratory incubation (Van de Velde et al., 2023). Is it possible that the difference between these two methodologies resulted in the shift of data points away from the $\log_{10}k$-$\log_{10}t$ anoxic trendline?

       - Was there any possible process that could interfere with the measurement of $^{35}S$ post-incubation (e.g., partitioning of consumed $^{35}SO_4^{2-}$ to $H_2S$ (g) and $FeS_2$ (solid))?

       - Was there any possible mechanism that can reduce or recycle $^{35}SO_4^{2-}$ without the need to consume OC?

       - Was there any new production of OC due to sulfate respiration (e.g., production of new microbial biomass) so that changes in sedimentary OC content does not directly followed 2:1 stoichiometry (Equation 4)?

       - Have the authors tried calculating k-values from changes in OC concentration with depth and time (using Katsev and Crowe (2015) method), without using SRR? Did they yield the same or different k-values?

**Line 206:** add ")" to "...Mayer, 1994b))".

**Line 225 (Figure 2):** Since the paragraph below discussed changes in OC loading with depth, I'd suggest the authors to improve this figure by showing the depth information on the plot. The authors may adjust the color of data point to represent depth (e.g., lighter color = surficial, darker color = deep), or they may just simply label the depths above every data point.

**Line 240:** Apart from intense OC mineralization, is there any alternative mechanism that potentially resulted in downcore decrease in OC loading? For example, was there any temporal change in mineral input to the Baltic sea? Was there any historical record of increasing light availability or

increasing nutrient input to the Baltic sea which could accelerate the production of autochthonous OC in recent years? Increasing OC production rate could result in higher OC:SSA in younger surficial sediment layers, compared to the older deeper sediments.

**Line 276-278:** "The lack of major rivers in this part of the Baltic Sea could explain the limited amount of OC-Fe found across the WGB sediments, supporting the hypothesis that $Fe_R$ preferentially binds terrestrial organic matter"

  It's not that $Fe_R$ preferentially binds with terrestrial or marine OC. The literature in this list observed that $Fe_R$ is preferrentially associated with terrestrial "vascular plant" OC, due to the abundance of reactive functional groups (such as phenols and carboxylic) in vascular plant materials. In the current study, molecular structure of OC is not the main factor controlling whether FeR will preferentially bind with terrestrial or marine OC. Vascular plant-OC is nearly absent in Baltic sea sediments while the majority of OC in is made of marine or freshwater POC (Fig. S1). Hence, the input of $Fe_R$ and OC to the Baltic may occur through separate processes. For example, land-derived $Fe_R$ may enter the Baltic Sea via overland flow while POC may be formed in-situ within water column. In this case, pre-formed $Fe_R$ was unable to react with newly formed OC.

  This leads to an important discussion that "while allochthonous land-derived OC-$Fe_R$ can survive anoxic conditions in marine sediments over millennia (Faust et al., 2021), the anoxic conditions inhibit autochthonous formation of new OC-$Fe_R$, even though the Baltic Sea received excessive input of both $Fe_R$ and OC from separate mechanisms".

**Line 283:** Replace "clay" with "mineral surface". There is no direct measurement of clay-bound OC in the current study.

**Line 285-286:** "Hence, our results suggest that ***limited physical protection reduces the importance of $O_2$ for OC mineralization***, as OC remains easily accessible for microbes, in this case, sulphate reducers"

  Linguistically, I think "***limited physical protection reduces the importance of $O_2$ for OC mineralization***" is not an appropriate statement. I think $O_2$ is still an important factor for OC mineralization in the condition of limited physical protection. In other words, unprotected OC will be degraded faster in the aerobic conditions (due to the ease of electron transfer etc.), compared to the anaerobic conditions.

  I'd suggest the authors to restructure this sentence to "***the absence of $O_2$ does not necessarily result in deceleration of OC decomposition, as we observed high SRR in anoxic Baltic Sea sediments. This significant loss of OC in anaerobic condition is likely due to limited physical protection of OC by minerals***".

---

## Author Comment (AC1)

**Review 1**

EGUsphere-2025-3020 Manuscript Review: Limited physical protection leads to high organic carbon reactivity in anoxic Baltic Sea sediments

Summary

To investigate the importance of mineral protection on OC reactivity in anoxic conditions, sediment cores were collected from the anoxic Baltic Sea. Their OC%, C/N ratio, $\delta^{13}C$, sulfate reduction rate, $^{210}Pb$ age, OC loading, and OC-FeR were determined. The downcore increase in OC% and SRR suggested intensive sulfate reduction and OC remineralization, suggesting high OC reactivity despite the anoxic conditions. High OC loading, with downcore decrease in OC/SAA, suggests inadequate mineral surface area for OC stabilization. Low abundance of OC-FeR indicated limited protection of OC by FeR minerals. In summary, the current study demonstrates that when mineral protection of OC is nearly absent, oxygen depletion alone could not decrease OC mineralization rate.

Major comments

Overall, this manuscript is well-written. The data set presented in the current study is small and simple, yet it addresses an important research question in the chemical oceanography field.

Therefore, I believe that the current manuscript contains a global significance that could be beneficial for further scientists who aim to investigate the drivers of OC preservation. Most of the methods and discussions are logical. I'd suggest this manuscript a minor revision, following my comments below.

Specific comments

Q1: Line 23-24: " Overall, these results suggest that the WGB sediments receive large amounts of OC relative to the supply of mineral particles, far exceeding the potential for OC physical protection"

This statement is true for OC/SAA; however, it might not be right for OC-FeR since the authors discussed that FeR availability is not the major factor controlling OC-FeR association (Line 267-273). Alternatively, I think the lack of OC-FeR in Baltic Sea is likely due to the separate input of FeR (from land) and OC (from freshwater/marine water column). See my comments in line 276-278.

A1: We agree with the reviewer that the $Fe_R$ availability alone does not control the extent of OC-Fe association. However, the SSA of these sediments partly reflects that of the Fe minerals dissolved through the CBD extraction. Our previous investigations in Swedish fjords (Placitu et al., 2024) suggested that riverine inputs can deliver preformed OC-Fe to the sediments. In contrast, the WGB receives limited riverine input, and the OC pool is dominated by autochthonous marine production rather than terrestrial OM. This likely reduces the potential for stable OC-Fe association to form.

Revised text in section 3.3: "The low %OC-Fe found in WGB1 and WGB2 potentially reflects both the decoupled delivery of $Fe_R$ and OC and the absence of substantial terrestrial inputs of $Fe_R$ that preferentially bind terrestrial organic matter. In the WGB, $Fe_R$ is likely to enter the Baltic Sea via surface runoff and is redistributed via sediment resuspension as described in Nilsson et al., (2021), while most OC consists of autochthonous marine production (Figure S1 in Supplementary information) which likely limits the formation of OC-Fe associations. This decoupling

of OC and $Fe_R$ inputs in WGB could explain the limited amount of OC-Fe found across the WGB sediments, supporting the hypothesis that $Fe_R$ preferentially binds terrestrial organic matter (Dicen et al., 2019; Linkhorst et al., 2017; Riedel et al., 2013; Salvadó et al., 2015; Shields et al., 2016; Sirois et al., 2018; Wang et al., 2019; Zhao et al., 2018, 2023). »

Placitu, S., van de Velde, S. J., Hylén, A., Hall, P. O. J., Robertson, E. K., Eriksson, M., Leermakers, M., Mehta, N., and Bonneville, S.: Limited Organic Carbon Burial by the Rusty Carbon Sink in Swedish Fjord Sediments, Journal of Geophysical Research: Biogeosciences, 129, e2024JG008277, https://doi.org/10.1029/2024JG008277, 2024.

Q2: Line 74: Add "," after "(Andrén et al., 2000)".

A2: Done

Q3: Line 82: In my opinion, "particulate organic carbon (POC)" is not an appropriate terminology here. Many literatures defined POC as suspended OC particles within water column, which has not deposited and has not been incorporated into sedimentary storage yet (see reviews in Bianchi (2007)). Another set of literature defined POC as a subset of bulk sedimentary OC that is made of fragmented plant litter with low density and large particle size (e.g., Cotrufo et al., 2019). To my understanding, the current study measured OC content in bulk sediment samples. Therefore, to avoid confusion, I'd suggest the authors to change the term "particulate organic carbon (POC)" to "sedimentary organic carbon (SOC)".

Bianchi, T. S. (2007). Biogeochemistry of estuaries. Oxford University Press.

Cotrufo, M. F., Ranalli, M. G., Haddix, M. L., Six, J., & Lugato, E. (201S). Soil carbon storage informed by particulate and mineral-associated organic matter. Nature Geoscience, 12(12), S8S-SS4.

A3: We agree with the proposed change and have replaced "particulate organic carbon (POC)" with "sedimentary organic carbon (SOC)" throughout the manuscript.

Q4: Line 150: What's about OC mineralization by reduction of $NO_3^-$ to $NH_4^+$, or reduction of $Fe^{3+}$ to $Fe^{2+}$?

A4: In persistently anoxic sediments, the contribution of denitrification, DNRA, manganese and iron reduction to organic carbon mineralization is negligible compared to sulphate reduction, given the dominance of sulphate in the porewater geochemistry. If these pathways were quantitatively important, the total reactivity would be even higher than our estimates, meaning our current mineralization rates are likely underestimated.

We included an additional sentence in the main text "These sediments are anoxic, unaffected by bioturbation, and sulphate $SO_4^{2-}$ is not depleted, so one can assume that all OC mineralization results from sulphate reduction. Contributions from denitrification, DNRA, manganese and iron reduction are negligible; however, it is important to note that the reported rates may slightly underestimate total reactivity"

Q5: Line 164-165: "...inputs, reflected in the $\delta^{13}C$ signature and C/N ratios of these sediments, highlighting the presence of 'fresh' material"

$\delta^{13}C$ and C/N ratios are proxies of OC sources, not their freshness. I do agree with the authors that the principal sources of OC in WGB cores are freshwater and marine planktons.

However, these proxies do not reflect whether these OC are made of 'fresh' plankton-derived OC, or 'degraded' plankton-derived OC. To determine whether these OC are fresh or degraded, the authors need additional proxies such as $\Delta^{14}C$ (age of OC) or biomarkers such as stanol/sterol ratios (higher stanol reflected more degraded OC).

A5: We agree with the comment, the word fresh was modified to ''autochthonous freshwater and marine material''

Q6: Line 182-183: To my understanding, Katsev and Crowe (2015) calculated k-values from changes in concentration of C with depth and time, using the equation $k = -\frac{U}{C}\frac{dC}{dX}$. In contrast, the current study calculated k-values from SRR, through the measurement of 35S radiotracer during laboratory incubation (Van de Velde et al., 2023). Is it possible that the difference between these two methodologies resulted in the shift of data points away from the $\log_{10}k$-$\log_{10}t$ anoxic trendline?

A6: We disagree with the reviewer on that point. Katsev and Crowe (2015) specify clearly that the use of burial velocity (U=dx/dt) is when dating of the sediment was not available. Where dating was available (which is our case here), they explicitly mention that k was directly calculated as k= $-C^{-1}$ dC/dt. As such, Katsev and Crowe dataset already contains k values derived from both expressions.

Q7 Was there any possible process that could interfere with the measurement of $^{35}S$ post-incubation (e.g., partitioning of consumed $^{35}SO_4^{2-}$ to $H_2S$ (g) and $FeS_2$ (solid))?

A7: The use of radioactive $^{35}S$ tracer (Jørgensen, 1978) is a well-established method to quantify sulphate reduction rates in sediments. To terminate incubations, zinc acetate (ZnAc) is added, which immediately precipitates all dissolved sulphide (including radiolabeled products) as insoluble ZnS. This prevents loss of volatile $H_2S$ and simultaneously halts microbial activity. Reduced sulphur species that have already reacted further into solid phases—such as FeS, $FeS_2$ and elemental sulphur—are subsequently recovered using the chromium reduction method (Canfield et al., 1984), in which these species are quantitatively converted to $H_2S$ and trapped again as ZnS (by adding ZnAc). All the ZnS formed is then titrated ensuring that both dissolved and solid-phase pools of reduced S are accounted for. Therefore, while partitioning of reduced sulphur into multiple pools does occur, the combined use of ZnAc fixation and chromium reduction ensures reliable SRR quantification.

Jørgensen BB. 1978 A comparison of methods for the quantification of bacterial sulfate reduction in coastal marine sediments: I. measurement with radiotracer techniques. Geomicrobiol. J. 1, 11-27. (doi:10.1080/01490457809377721)

Canfield DE, Raiswell R, Westrich JT, Reaves CM, Berner RA. 1986 The use of chromium reduction in the analysis of reduced inorganic sulfur in sediments and shales. Chem. Geol. 54, 149-155. (doi:10.1016/0009-2541(86)90078-1)

Q8: Was there any possible mechanism that can reduce or recycle $^{35}SO_4^{2-}$ without the need to consume OC?

A8: There are potentially other processes e.g. chemolithoautotrophic sulphur oxidizers (using reduced sulphur with nitrate) or the various disproportionation reactions that might potentially interfere with the $^{35}SO_4^{2-}$ incubations (without OC oxidation) identified in Baltic Sea sediments. However, we consider these metabolisms to be negligible relative to sulphate reduction. Indeed, van de Velde (2023) modelling shows a very good agreement between measured and modelled $SO_4^{2-}$ and DIC concentrations in WGB cores suggesting that sulphate reduction alone account for most of the OC mineralization and $SO_4^{2-}$ concentration variation at depth.

Q9: Was there any new production of OC due to sulfate respiration (e.g., production of new microbial biomass) so that changes in sedimentary OC content does not directly follow 2:1 stoichiometry (Equation 4)?

A9: Sulphate reducers can divert a small fraction of the consumed OC into anabolic reactions and new biomass, but this C assimilation is typically very low, on the order of a few percent max. of C substrate depending on the compounds (Stolyar et al., 2007; Londry & Des Marais, 2003; Pellerin et al.,2020). Thus, this assimilated fraction does not alter significantly the 2:1 stoichiometry.

Stolyar, S., Van Dien, S., Hillesland, K. L., Pinel, N., Lie, T. J., Leigh, J. A., & Stahl, D. A. (2007). *Metabolic modeling of a mutualistic microbial community*. *Molecular Systems Biology*, 3:92. https://doi.org/10.1038/msb4100131

Pellerin, A., Antler, G., Marietou, A., Turchyn, A. V., & Jørgensen, B. B. (2020). *The effect of temperature on sulfur and oxygen isotope fractionation by sulfate reducing bacteria (Desulfococcus multivorans)*. *FEMS Microbiology Letters*, 367(9), Article fnaa061. https://doi.org/10.1093/femsle/fnaa061 (Pure)

Londry, K. L., Jahnke, L. L., & Des Marais, D. J. (2004). *Stable carbon isotope ratios of lipid biomarkers of sulfate-reducing bacteria*. *Applied and Environmental Microbiology*, 70(2), 745-751. https://doi.org/10.1128/AEM.70.2.745-751.2004

Q10: Have the authors tried calculating k-values from changes in OC concentration with depth and time (using Katsev and Crowe (2015) method), without using SRR? Did they yield the same or different k-values?

A10: van de Velde et al., 2023 modelled OC respiration rate at these sites using the downcore OC profile (as in Katsev and Crowe, 2015) and with $SO_4^{2-}$ reduction rate and showed that they agree well (see Fig. 3 in van de Velde, 2023).

Q11: Line 206: add ")" to "...Mayer, 1994b))".

A11: Done.

Q12: Line 225 (Figure 2): Since the paragraph below discussed changes in OC loading with depth, I'd suggest the authors to improve this figure by showing the depth information on the plot. The authors may adjust the color of data point to represent depth (e.g., lighter color = surficial, darker color = deep), or they may just simply label the depths above every data point.

A12: We agree with this suggestion and improved the Figure 4 following the recommendation of the reviewer, i.e. with a colour code in each data point denoting depth.

Q13: Line 240: Apart from intense OC mineralization, is there any alternative mechanism that potentially resulted in downcore decrease in OC loading? For example, was there any temporal change in mineral input to the Baltic sea? Was there any historical record of increasing light availability or increasing nutrient input to the Baltic sea which could accelerate the production of autochthonous OC in recent years? Increasing OC production rate could result in higher OC: SSA in younger surficial sediment layers, compared to the older deeper sediments.

A13: The reviewer is right that there has been an acceleration of the production of autochthonous OC in large parts of the Baltic Sea in recent decades due to eutrophication. Indeed, the Baltic Sea has experienced major eutrophication during the 20th century (e.g., Conley et al. 2009), linked to nutrient enrichment, which likely increased autochthonous OC production and could contribute to higher OC:SSA values in more recent sediments relative to deeper, older layers. While resolving these temporal dynamics in WGB directly is out of the scope of the study, the observed downcore decline in OC:SSA is consistent with progressive degradation and mineralization of OC, as also suggested by high sulphate reduction rates.

Revised text: "The very high OC loadings observed in the WGB surface sediments most likely reflect a combination of intense autochthonous OC inputs, stimulated by eutrophication, together with a limited role of mineral surfaces in constraining OC accumulation." [...] "Enhanced nutrient loading and primary production in the Baltic Sea during the 20th century (Conley et al., 2009) likely contributed to these elevated OC:SSA values in surficial sediments, as increased deposition of autochthonous OC on particle surfaces was not supported by a proportional increase in mineral inputs."

Q14: Line 276-278: "The lack of major rivers in this part of the Baltic Sea could explain the limited amount of OC-Fe found across the WGB sediments, supporting the hypothesis that FeR preferentially binds terrestrial organic matter"

It's not that FeR preferentially binds with terrestrial or marine OC. The literature in this list observed that FeR is preferentially associated with terrestrial "vascular plant" OC, due to the abundance of reactive functional groups (such as phenols and carboxylic) in vascular plant materials. In the current study, molecular structure of OC is not the main factor controlling whether FeR will preferentially bind with terrestrial or marine OC. Vascular plant-OC is nearly absent in Baltic sea sediments while the majority of OC in is made of marine or freshwater POC (Fig. S1).

Hence, the input of FeR and OC to the Baltic may occur through separate processes. For example, land-derived FeR may enter the Baltic Sea via overland flow while POC may be formed in-situ within water column. In this case, pre-formed FeR was unable to react with newly formed OC.

This leads to an important discussion that "while allochthonous land-derived OC-FeR can survive anoxic conditions in marine sediments over millennia (Faust et al., 2021), the anoxic conditions inhibit autochthonous formation of new OC-FeR, even though the Baltic Sea received excessive input of both FeR and OC from separate mechanisms".

A14 : We thank the reviewer for pointing out the need to clarify the factors controlling OC-Fe associations in the Baltic Sea sediments. We agree that the molecular structure of OC is not the main factor controlling whether FeR will preferentially bind with terrestrial or marine OC, since vascular plant-derived OC is nearly absent and the majority of OC in Baltic sediments is marine SOC (Fig. S1). In response to the comment, we revised the text (see below) to clarify that the limited OC-Fe observed in the WGB sediments potentially reflects the decoupled delivery of $Fe_R$ and OC. Specifically, $Fe_R$ may be delivered via land-derived inputs (e.g., runoff and sediment resuspension), while SOC is largely produced *in situ* in the water column. This explanation aligns with previous observations that allochthonous OC-Fe complexes can survive over millennia, while autochthonous formation of OC-Fe is limited under anoxia (Faust et al., 2021). We believe this revision addresses the reviewer's concern and provides a more comprehensive interpretation of the mechanisms controlling OC-Fe formation in the Baltic Sea sediments.

Revised text: "The low %OC-Fe found in WGB1 and WGB2 potentially reflects both the decoupled delivery of $Fe_R$ and OC and the absence of substantial terrestrial inputs of $Fe_R$ that preferentially bind terrestrial organic matter. In the WGB, $Fe_R$ likely enter the Baltic Sea via surface runoff and is redistributed via sediment resuspension, while most OC consists of autochthonous marine production (Fig. S1 in Supplementary information) which likely limit the formation of OC-Fe associations. The decoupling of OC and $Fe_R$ inputs in WGB could explain the limited amount of OC-Fe found across the WGB sediments, supporting the hypothesis that $Fe_R$ preferentially binds terrestrial organic matter (Dicen et al., 2019; Linkhorst et al., 2017; Riedel et al., 2013; Salvadó et al., 2015; Shields et al., 2016; Sirois et al., 2018; Wang et al., 2019; Zhao et al., 2018, 2023).

Q15: Line 283: Replace "clay" with "mineral surface". There is no direct measurement of clay-bound OC in the current study.

A15: Agree, corrected.

Q16: Line 285-286: "Hence, our results suggest that limited physical protection reduces the importance of $O_2$ for OC mineralization, as OC remains easily accessible for microbes, in this case, sulphate reducers"

Linguistically, I think "limited physical protection reduces the importance of $O_2$ for OC mineralization" is not an appropriate statement. I think $O_2$ is still an important factor for OC mineralization in the condition of limited physical

protection. In other words, unprotected OC will be degraded faster in the aerobic conditions (due to the ease of electron transfer etc.), compared to the anaerobic conditions.

I'd suggest the authors to restructure this sentence to "the absence of $O_2$ does not necessarily result in deceleration of OC decomposition, as we observed high SRR in anoxic Baltic Sea sediments. This significant loss of OC in anaerobic condition is likely due to limited physical protection of OC by minerals".

A16: We thank the reviewer for this clarification. We agree that our initial phrasing could be misinterpreted as suggesting that $O_2$ is not important for OC mineralization, which is not what we intended. As the reviewer points out, oxygen remains the most efficient electron acceptor and OC will be degraded faster under oxic than anoxic conditions. Our intended message was that, in the WGB sediments, the absence of $O_2$ does not necessarily slow down OC decomposition, since high SRR indicate that "unprotected" OC is still readily available for anaerobic degradation. We revised the ending of the conclusion accordingly:

Revised text:" Together with consistently high sulphate reduction rates across the basin, these results suggest that anoxic conditions in the WGB do not necessarily result in a slowdown of OC mineralization. Instead, the significant degradation of OC under anaerobic conditions is likely facilitated by the limited physical protection of OC by mineral surfaces, which keeps a large portion of OC accessible to sulphate reducers. "

---

## Author Comment (AC2)

**Review 2**

**Limited physical protection leads to high organic carbon reactivity in anoxic Baltic Sea sediments**

**General Assessment:**

Q1: This manuscript addresses a timely and relevant topic in sedimentary organic carbon dynamics under varying redox conditions. While the subject is of clear interest to the biogeoscience community, I find that the study, in its current form, falls short in several key areas that significantly limit its impact and interpretive strength.

Most critically, the data presented are sparse and inconsistently applied across stations, with core analytical components (e.g., $\delta^{13}C$, Fe-bound OC) performed only at select sites, and important parameters such as $\Delta^{14}C$ not included at all. This uneven dataset complicates cross-site comparisons and weakens the manuscript's overarching conclusions.

A1: We agree that a full comparison of OC–Fe associations across all stations would further strengthen the dataset. However, OC-Fe extractions are analytically demanding, and therefore we focused this analysis on two contrasted sites (WGB1 and WGB2) that represent key environmental discrepancies in the WGB basin (i.e., hypoxic vs. anoxic bottom waters). This targeted approach allowed us to capture the main variability relevant to our study objectives while keeping the experimental work tractable.

While we did not conduct OC–Fe extractions at all five stations, we complemented this targeted dataset with a broader suite of measurements (OC content, $Fe_R$ concentrations, SRR, C stable isotopes, C/N ratio etc), which provide consistent evidence across stations. Importantly, as mentioned above the two sites chosen for OC–Fe extractions bracket the range of conditions observed in the WGB, and thus we believe they are representative of the processes controlling OC preservation in this system. Regarding $\Delta^{14}C$, we agree that radiocarbon measurements would add valuable constraints on OC sources, age and turnover. However, this was beyond the scope of the present study. We have revised the manuscript text to clarify these limitations in the conclusion as being supported primarily by the contrasting conditions at WGB1 and WGB2, supplemented by supporting evidence from the other sites.

Revised text: "Sediments in the WGB underly hypoxic and anoxic bottom waters, yet the reactivity of sedimentary OC is in line with previous estimates from oxic settings. Detailed analyses at two representative sites (WGB1 and WGB2) show limited potential for physical protection of OC by adsorption on mineral surfaces and association with Fe oxides in these sediments, demonstrated by high OC loadings and low amounts of OC-Fe associations. Together with consistently high sulphate reduction rates across the basin, these results suggest that anoxic conditions in the WGB do not necessarily result in a slowdown of OC mineralization. Instead, the significant degradation of OC under anaerobic conditions is likely facilitated by the limited physical protection of OC by mineral surfaces, which keeps a large portion of OC accessible to sulphate reducers. While our targeted dataset does not capture the full extent of spatial heterogeneity of the WGB, it nevertheless supports the view that mineral protection, rather than redox state alone, constitutes a critical control on the long-term burial of OC in marine sediments."

Several of the interpretations appear speculative and are not strongly supported by the data. Furthermore, the discussion lacks depth in many areas, especially regarding key observations in the POC and SRR profiles, and the potential mineralogical and microbial controls on OC preservation (expanded below in "comments").

The manuscript would benefit from a more robust theoretical framework and a clearer connection between the presented data and the cited literature. Some central concepts—such as mineral phase protection, particle

shuttling, and OC-Fe associations—are mentioned but not fully developed or critically examined in light of the actual data.

**Comments**:

Q2: Line 37: "... role in regulating the sediment biogeochemistry and faunal community, which in turn influence OC cycling??" As currently written, the linkage to OC remineralization is vague and needs to be explicitly connected to the broader context introduced in the preceding sentence.

A2: We have clarified the link between $O_2$ availability, sediment biogeochemistry, faunal activity, and OC cycling. The revised sentence (line 37) now states that $O_2$ not only regulates sediment redox processes and benthic fauna but also influence the OC mineralization and OC burial (e.g., via bioturbation, bioirrigation, and redox-dependent microbial metabolisms).

Q3: Lines 39-43: These paragraphs read awkwardly. The authors initially suggest that anoxia promotes OC preservation but then shift to discussing the role of oxygen in anabolism. Although these concepts are related, the logic and flow need clarification.

A3: We rephrased this sentence to clarify our reasoning.

Revised text: It is often reported that sediments deposited under anoxic bottom waters tend to exhibit higher apparent OC preservation. This pattern is generally interpreted not as an active preservation mechanism of anoxia, but rather as the consequence of $O_2$ strongly enhancing OC mineralization when present. Several processes explain this : (i) aerobic respiration yields more free energy than anaerobic pathways, leading to faster degradation (LaRowe and Van Cappellen, 2011), (ii) certain complex organic molecules, such as lignin, can be more efficiently degraded via oxidative enzymes available only under aerobic conditions (Megonigal et al., 2003; Burdige, 2007; Canfield, 1994), (iii) aerobic organisms are capable of degrading OM completely to $CO_2$ (via the Krebs cycle) while mineralization under anoxic condition proceeds through a multi-step anaerobic food chains involving syntrophic interactions among specialized consortia which makes the degradation slower and less efficient (Arndt et al., 2013); (iv) oxygenated sediments support macrofauna that can directly or indirectly stimulate mineralization through bioturbation and bioirrigation and prevent accumulation of reduced toxic products such as $H_2S$ (Kristensen et al., 1992; Aller and Aller, 1998; Papaspyrou et al., 2007; van de Velde et al., 2020).

Q4: Also, the sentence about complex molecules being more easily degraded via aerobic pathways lacks detail. What specific compounds are meant here?

A4: See above A3, we included ''such as lignin'' in lignin' 'in line 44 ''.

Q5: The statement regarding certain compounds requiring oxygen for degradation would benefit from examples and context. How representative are these compounds relative to bulk OC? And the distinction between points (ii) and (iii) is unclear and may be redundant. Please re-word this.

A5: See above A3, we rephrased this whole section. We now cite lignin as an example of compounds being more readily degraded in oxic environment. Lignin generally constitutes a small proportion (<10%) of bulk OC in most coastal sediments, its degradation nevertheless exemplifies the role of oxygen in breaking down complex, terrestrially derived compounds.

Q6: Line 43: What about discussing the role of microbial consortia in OC remineralization? This will strengthen your arguments

A6: We agree that the role of microbial consortia is important to understanding OC remineralization. We have therefore added a statement emphasizing that aerobic microbial communities are generally more versatile and efficient in degrading a wide range of organic compounds, whereas anaerobic degradation often depends on slower, syntrophic interactions among specialized consortia (see A3 – point iii). This addition complements our discussion of the thermodynamic, enzymatic, and faunal mechanisms underlying enhanced aerobic mineralization.

Q7: Lines 110-112: Please provide more details on the standards used for stable carbon isotope analyses, as well as QA/QC measures. Information on how linearity, drift, and reproducibility were assessed would strengthen the methodological section.

A7: We included additional details in the text as suggested "The samples were then dried under an infrared lamp and transferred into a tin capsule before being loaded into on an elemental analyser (Sercon Europa EA-GSL; Sercon Ltd., Crewe, UK) coupled to an isotope ratio mass spectrometer (IRMS; Sercon 20-22) at the ISOGOT facility (Department of Earth Sciences, University of Gothenburg). A certified reference material (high organic content sediment standard; Elemental Microanalysis, UK) was used as an internal standard throughout analysis, approximately every 10-12 samples. This standard has a certified isotope value of $\delta^{13}C$ -26.27 ‰. Drift was corrected relative to the standard and reproducibility was determined from replicate standard analyses. Blanks were run at the beginning of each analytical run to verify negligible contamination from previous analyses.) Linearity was assessed during annual service using reference gases ($CO_2$)."

Q8: Line 113: Table SI 1 not displaying this, but Lalonde method.

A8: Thanks, we corrected citing the dataset in the supplementary information instead.

Q9: Line 123: Was there a rationale for using plastic Falcon tubes for OC analysis? These are generally not recommended due to potential contamination. Why were pre-combusted glass vials not used?

A9: We thank the reviewer for this comment. While it is true that pre-combusted glass vials are often recommended to avoid contamination for low-carbon samples (e.g. for DOC analysis), in our study the sediments contain high concentrations of organic carbon (tens of mg C $g^{-1}$), making any potential contribution from plastic Falcon tubes negligible in comparison. We therefore consider the use of plastic tubes appropriate for the OC analyses performed, as the sedimentary OC overwhelms any minor carbon contamination from the Falcon tubes.

Q10: Lines 127-128: More information is needed regarding cleaning protocols for filters and collection containers.

A10: All filters and collection containers used during the extraction were single-use, and therefore no additional cleaning was performed prior to use. To check any contamination from filters or tubes, blank samples were included and analyzed with the sediment samples. These blanks confirmed that any contribution from the extraction materials was negligible.

Q11: Lines 155-157: You acknowledge that $^{210}$Pb-derived OC age estimates do not capture all relevant degradation pathways, yet no further discussion is provided. Consider elaborating on how this methodological limitation may influence your conclusions, and whether complementary proxies were considered.

A11: Indeed, we acknowledge that $^{210}$Pb-derived sediment ages reflect the time since deposition at the sediment surface and therefore underestimate the true age of OC and thus the degradation it may have experienced prior to deposition on the sediment surface. However, in the WGB sediments, the majority of OC is autochthonous, i.e. produced in the water column before deposition. As a result, the potential bias in OC age due to prior degradation is likely minimal, and the $^{210}$Pb-derived ages provide a reasonable approximation for assessing sedimentary OC reactivity. However, in the case that age would be drastically underestimated, this would only shift the log k vs log t

trend further to the right in Fig. 3 and hence reinforce our argument that the OC in WGB is degraded at fast rates. Complementary proxies, such as $\delta^{13}C$ and C/N ratios (Fig. S1) were also considered to evaluate OC sources (and to a certain extent its lability), supporting the interpretation that OC in these sediments is highly reactive and that its apparent age aligns well with depositional history.

Q12: Lines 160-165: The statement that POC profiles are "similar" across sites oversimplifies the data. For example:

* WGB1 has lower surface concentrations and a more gradual decline with depth than WGB2 and WGB3.

*WGB4 shows an apparent increase below 5 cm.

These contrasting profiles merit discussion. Avoid generalizations that obscure site-specific differences.

A12: We thank the reviewer for this observation. We agree that there are site-specific differences in the SOC profiles and have revised the text to better reflect this. While the overall trend across stations shows a decrease of OC with depth, the magnitude and shape of the profiles indeed differ. In particular, the relatively low surface OC concentrations at WGB1 and the deeper increase observed at WGB4 may reflect local depositional and transport processes. Recent work has shown that particle shuttling from shallow to deeper parts of basins in the Baltic Sea can redistribute organic matter and create spatial heterogeneity in OC distribution and burial (Nilsson et al., 2021). We have added this perspective to the discussion, emphasizing that variability in POC profiles across sites likely reflects both local mineralization dynamics and lateral particle transport, rather than being purely a function of degradation and its change with sediment depth at each site.

Revised text: "The sedimentary organic carbon (SOC) profiles (Figure 2, A-E) generally decrease with depth at all stations, but with distinct site-specific trends. At WGB2 and WGB3, the deepest stations (Table 1), surface SOC contents reach up to 20 wt%, while WGB1 shows lower surface values and a more gradual decline. At WGB4, an increase at ~5 cm depth may suggest a potential, localized input of refractory OC (possibly of terrestrial origin). Elevated SOC concentrations in the Baltic Sea generally stem from cyanobacteria, diatoms, and other phytoplankton blooms induced by nutrients inputs, as indicated by $\delta^{13}C$ signatures and C/N ratios of WGB1 and WGB2 sediments (Figure S1 in Supplementary information), reflecting autochthonous freshwater and marine inputs. At WGB1, progressive $\delta^{13}C$ depletion with depth and increasing C/N indicate selective microbial degradation of labile, N-rich compounds. In contrast, the absence of a trend with depth in $\delta^{13}C$ at WGB2 suggests that OC burial at this station is more strongly influenced by lateral OC inputs. This observation is consistent with the "shuttling" of particulate material described by Nilsson et al., (2021) where repeated cycles of resuspension-redeposition transport particle from shallow, erosive areas (such as WGB1) to deep accumulation areas (such as WGB 2). In addition to this transport, since the mid 1900's, enhanced nutrient loading in the central Baltic Sea has increased OC inputs, while damming of major rivers reduced sediment delivery and limited dilution of autochthonous OC, contributing to the heterogeneity of SOC profiles in WGB (Emeis et al., 2000; Leipe et al., 2011). »

Q13: Line 165: Only $\delta^{13}C$ data from WGB1 and WGB2 are shown, yet the sentence discusses WGB2 and WGB3. Consider presenting $\delta^{13}C$ data from all stations for consistency and interpretive strength.

A13: We clarified in line 165 that the $\delta^{13}C$ and C/N data discussed are specifically from WGB1 and WGB2 sediments as these were the only stations where such measurements were performed.

Q14: Lines 164-169: Vague statement.

A14: We clarified this section in our answer A12.

Q15: Lines 170-173: As mentioned for POC, the discussion regarding SRR is brief and does not adequately address key variations between stations.

For example: there are striking differences in SRR between WGB2 and 3, the latter displaying much higher values, sharply decreasing with depth in the top 5 cm, while the WGB2 has a relatively uniform SRR. How does this and the fact that their POC profiles are similar do not grant any further discussion?

Similar remarks can be made for WGB4 and WGB5, which display similar SRR but contrasting POC depth patterns. **These observations deserve further discussion beyond a simple correlation with OC reactivity.**

A15: We agree that the SRR depth profiles reveal site-specific contrasts that may go beyond a simple correlation with OC reactivity. In the revised manuscript, we now highlight that WGB3 displays much higher SRR values in the surface layer compared to WGB2, despite both stations exhibiting broadly similar SOC profiles. This suggests that factors other than bulk SOC availability — such as differences in the composition of the organic matter and/or microbial consortia and structure — may explain the higher SRR and their sharp decline at WGB3. With regard to WGB4 and WGB5, however, we note that their SRR depth profiles are in fact quite comparable and the apparent contrast in SOC depth trends is much less pronounced than suggested. Nevertheless, we now emphasize that these overall cross-site comparisons support the conclusion that OC accessibility and quality, not just total OC content, are the key regulators of sulphate reduction in these sediments.

Revised text: "The SRR depth profiles (Figure 2, F–J) show generally elevated activity in the surficial sediments, consistent with the distribution of sedimentary organic carbon. Stations WGB2 and WGB3 exhibit the highest SRR values, but with contrasting depth trends: WGB3 displays markedly higher surface rates that decline steeply within the upper 5 cm, whereas WGB2 shows more moderate but relatively uniform SRR throughout the upper layers. This difference, despite broadly similar SOC depth profiles, suggests that SRR is influenced not only by bulk OC content but also by others factors such as organic matter degradability, microbial community and porewater geochemistry."

Q16: Line 174: The term "OC age" is potentially misleading, especially in the absence of radiocarbon data. This may lead to misinterpretation of your findings.

A16: We thank the reviewer for pointing out the potential ambiguity of the term "OC age." In our study, the ages are not based on radiocarbon dating but are inferred from [210]Pb-derived sediment chronologies, which provide estimates of the time since deposition at the sediment surface. To avoid misinterpretation, we have revised the terminology throughout the text to "[210]Pb-estimated age" instead of "OC age."

Q17: Line 182: Please use terminology consistently. "OC age" is not appropriate for [210]Pb-based estimates. The figure refers to "sediment age," which should be used throughout for clarity.

A17: See A16.

Q18: Lines 186: The explanation and analysis of this is quite superficial, there is little exploration of why some station plot toward one line and others to the anoxic line. Some samples are split in between. Without independent proxies of OC reactivity, the conclusion lacks sufficient support.

A18: We have clarified our interpretation in the revised manuscript. The variability in the positioning of our data relative to the oxic and anoxic reference lines can be explained by the station-specific SRR, which were used to derive the first-order decay rate constant (k). Stations plotting closer to the oxic line generally correspond to higher SRR values, reflecting faster OC turnover, whereas stations plotting closer to the anoxic line display lower SRR and hence lower apparent OC reactivity. This consistency between SRR patterns and the positioning of the *k* value

dataset supports the conclusion that differences in microbial activity, rather than redox conditions alone, govern the observed spread.

Revised text: Because large parts of the WGB are long-term anoxic, the *k* values would be expected to follow the reactivity-age relationship derived for anoxic sediments. Intriguingly, however, most of the *k* values fall closer to the oxic reactivity-age trend line (Figure 3). This spread can be explained by differences in SRR, which were used to calculate *k*: stations with higher SRR (e.g., WGB3) plot nearer to the oxic relationship, while those with lower SRR (e.g., WGB2) tend toward the anoxic trend. Our findings indicate that microbial activity, reflected in SRR, exert a stronger influence on OC reactivity than the redox setting itself. Note that the [210]Pb-estimated age of OC may underestimate the true age (see section 2.4) which would shift the data further to the right toward the oxic trendline.

Q19: Lines 185-190: This paragraph is somewhat tangential. The connection to the Vatsev study is not clearly explained, and the comment on zooplankton targeting reactive OC appears speculative. Consider focusing more directly on how your results relate to established degradation mechanisms.

A19: We disagree with the reviewer on this point. We consider this section central to place our results into the context of existing OC reactivity–age frameworks such as that of Katsev & Crowe (2015). The compilation of these authors contrasts oxic water column data with anoxic sediment data, and without this clarification, the comparison of our k values with their trends could be misleading. In our view, the discussion of particulate organic matter processing during sinking is not speculative as it is based on well-established degradation mechanisms described in Arndt et al. (2013) and Burdige (2007), who emphasize the preferential consumption of more reactive fractions in the water column and the delivery of comparatively less reactive OC to sediments. By including this explanation, we ensure that the reader understands why the WGB sediments, despite being deposited under anoxic conditions, exhibit high OC reactivity. It is not a direct oxygen effect but rather linked to the limited mineral protection of deposited OC in these sediments. We believe this clarification is essential for a robust interpretation of our results in relation to existing literature.

Q20: Lines 190-192: The cited literature primarily addresses mineral associations with terrestrial OC, which differs from your dataset dominated by marine-derived material. The relevance of these studies to your interpretations should be clarified.

A20: We would like to clarify that several of the references we cite (e.g. Mayer, 1994b; Keil., 1994; Li et al. 2017) are based on marine sediments and specifically address the mechanism of OC-mineral interactions in marine depositional settings. While some of the studies we also mention (e.g. Goni et al, 2008 and Blair and Aller, 2012) investigate interactions between terrestrial OC and sediment particles, the process of sorption to mineral surface and OC stabilisation with minerals are not restricted to terrestrial inputs. The cited references are thus directly relevant to our dataset, which is dominated by marine organic matter. To avoid any ambiguity, we explicitly made clarifications in the revised text to specify the settings of the reference we cite.

Added text: "While this mechanism has been well documented in terrestrial settings (Goni et al., 2008; Blair and Aller, 2012), where sorption provides protection to inherently more refractory compounds such as lignin-rich material, it has also been demonstrated in marine sediments (Mayer, 1994b; Keil et al., 1994; Li et al., 2017). In the latter case, sorption on mineral surfaces may help to stabilize more labile, phytoplankton-derived OC."

Q21: Lines 194-195: It would be important to contrast the lability of marine-derived OC from phytoplankton blooms with terrestrial OM. Much of the literature on mineral phase protection involves terrestrial inputs, which are not considered here.

A21: We agree that terrestrial and marine OC differ in composition and lability, with terrestrial OC generally being more refractory due to its high content of lignin and complex macromolecules, whereas marine-derived OC from phytoplankton blooms is comparatively more labile and more readily degradable. However, the mechanisms of sorptive protection and mineral association have been shown to operate with both terrestrial and marine OC. We have revised our text to acknowledge the difference in reactivity of terrestrial and marine OM. See A20.

Q22: Line 225: Figures 3 and 4 refer to "data per station," but it is unclear which depths or samples are included. Are all depths shown? Why are there more data points in Figure 3 than Figure 4, despite nearly identical captions? ("data are represented per station...").

A22: We have revised Figure 3 and 4 to improve clarity. In both Figures, symbol colours now indicate sample depth. Figure 3 includes all available data across stations ad depths, whereas Figure 4 is restricted to samples where both SOC and SSA were measured, which explains the smaller dataset.

Q23: Also, consider incorporating sediment depth into the figure using a color bar or symbol shading to help convey vertical structure in the data.

A23: Done, see A22.

Q24: Line 228: I cannot see this in the supplement available, also, why are not you showing all the depth in Figure 4? Which depth and which rationale was used to show some of the data there.

A24: Done see A22.

Q25: Line 230-240: The entire section here reads as a summary of disconnected concepts rather than a coherent interpretation. There's no discussion of how OC/SSA relates to SRR or OC%. We cannot assess why some surficial (circled) samples behave so different in the POC / SSA space, with WGB3 and 1 plotting below the OC/SSA= 10 line. This section would benefit from a more detailed analysis of the relationship between OC loading and measured SRR values. Currently, key patterns in the SRR data are not addressed, and their relevance to the OC/SSA trends remains unclear. For example, the surface sample at WGB3 exhibits the highest SRR among all stations, with a sharp decline downcore, whereas WGB1 shows much lower SRR values, yet it plots closer to the OC/SSA = 1 line. Similarly, WGB4 and WGB5 exhibit a marked decrease in OC loading with depth, while WGB2 displays a more subtle pattern despite having a comparable SRR profile. These contrasts suggest decoupling between OC reactivity and mineral association that deserve further exploration. A more integrated discussion connecting SRR profiles with OC/SSA trends is needed to support the interpretations in this section.

A25: We disagree with the assessment that this section is a collection of disconnected concepts. Our intent was to build a coherent interpretation linking OC/SSA values, SOC, and SRR to the extent of mineral protection in WGB sediments. To address the reviewer's concern, we have clarified these connections in the revised text. Specifically, we now emphasize (i) that surface samples with the highest OC/SSA also exhibit the highest SRR, consistent with elevated SOC availability, (ii) that the strong decline in OC/SSA with depth mirrors both the decrease in SOC and SRR, reflecting efficient mineralization, and (iii) that surficial samples at WGB1 and WGB3 plot below the OC/SSA = 10 line is likely due to lower surface SOC concentrations relative to mineral surface area, indicating site-specific differences in depositional inputs rather than inconsistencies in the overall pattern. These revisions ensure that the section more clearly integrates OC/SSA with the discussion of OC reactivity and mineral protection.

Revised text: "The very high OC loadings observed in the WGB sediments most likely reflect a combination of intense autochthonous OC inputs, stimulated by eutrophication, together with a limited role of mineral surfaces in constraining OC accumulation. OC loadings range from approximately 2 mg OC m$^{-2}$ in the hypoxic WGB1 to over 6

mg OC m$^{-2}$ at other stations, peaking at around 10 mg OC m$^{-2}$ in WGB4 (Figure 4). These values are comparable to those reported in upwelling regions or other settings with high productivity such as the Peruvian slope of Black Sea sediments (Mayer, 1994a). Enhanced nutrient loading and primary primary production in the Baltic Sea during the 20$^{th}$ century (Conley et al., 2009), likely contributed to these elevated OC/SSA values in surficial sediments, as higher fluxes of autochthonous OC accumulated on particles surfaces compounded by the reduced dilution from mineral inputs. With depth, OC loadings systematically decline across all stations, dropping below 1 mg C m$^{-2}$ beneath 10 cm and falling below the monolayer equivalent adsorption threshold of 0.86 mg C m$^{-2}$ at 15 cm (Mayer, 1994b). This downcore decrease parallels the sharp decline in both SOC concentrations and SRR (Figure 2), indicating that high surface loadings represent reactive organic pools that are efficiently degraded rather than being stabilized by mineral associations. Thus, the combined OC/SSA, SRR, and SOC patterns point to a system where microbial activity dominates over mineral protection in regulating OC burial in WGB sediments."

Q26: Lines 244-280: The entire section 3.3 is highly speculative. It brings up numerous concepts from literature without clearly linking them with study data. How you could confidently assess the role of physical protection by only accounting for Fe-OC; there are other mineral phases well known to aid preservation of OC, besides Fe-OC.

A26: We disagree with the assessment of the reviewer regarding section 3.3. This section is presenting original dataset of OC-Fe that is the basis of our discussion. Having said that, we agree that OC-Fe represent only one component of the broader suite of mineral phases that can stabilize organic matter. However, our interpretation does not rely solely on OC-Fe associations. In the preceding section, we analysed OC/SSA and OC loading relationships, which provide an integrated view of mineral-associated organic matter across all particle surfaces. These analyses showed extremely high OC loadings relative to SSA, indicating that mineral sorption overall plays a limited role in protecting OC in WGB sediments. To confirm the limited role of physical protection in the WGB, we used OC-Fe measurements, given the importance of Fe mineral phases to stabilize OC in many environments (e.g., Lalonde et al. 2012; Zhao et al. 2018). The very low %OC-Fe$_R$ values we measured (0-7%) compared to global averages (15-20%) reinforce the interpretation already suggested by our OC/SSA data, i.e. that mineral protection from Fe minerals or other phases, contributes only marginally to OC burial in WGB.

We significantly revised section 3.3 to make this connection explicit and to clarify that OC-Fe should not be viewed as the sole mechanism of protection, but rather as an additional line of evidence supporting our OC/SSA findings.

Revised text: "The very high OC loadings observed in WGB sediments (section 3.2) indicate that mineral surfaces overall play a limited role in preserving OC from degradation. This interpretation is reinforced by the very low fraction of OC bound to Fe$_R$ (Figure 5), compared to a global average for marine sediments of 15-20% (Lalonde et al., 2012; Longman et al., 2022; Zhao et al., 2018)." [....] Taken together, these results indicate that Fe minerals contribute only marginally to OC preservation in the WGB. While we cannot rule out that other mineral phases (e.g. clays, Al oxides) also provide sorption sites, the combination of high OC loadings and very low %OC-Fe strongly suggest that physical protection is very limited in WGB."

Q27: The analysis of Fe-bound OC is limited to only two stations and lacks interpretive depth.

A27: We addressed this comment in A1.

Q28: Is OC-Fe from WGB2 real, or an artifact of the extraction procedure? WGB2 appears to show erratic Fe-OC values (only 2 points have OC-Fe > 0) without clear depth trends, yet this is not discussed. In the manuscript only ranges are discussed, which indeed does not tell much.

A28: We acknowledge that the %OC-Fe$_R$ are low and do not display a systematic depth trend. However, we do not interpret these results as an artefact of the extraction method. The CBD extraction is a well-established method

for quantifying reactive Fe pools and associated OC (e.g. Lalonde et al., 2012; Faust et al., 2021; Barber et al., 2014; Yao et al., 2023; Mehra & Jackson, 1958). The few depths showing OC-Fe associations in WGB2 fall within the expected range for OC-Fe associations in marine sediments (albeit at the low end) (e. g see Longman et al,. 2022), suggesting that they are real signals rather than artefacts. The absence of clear downcore trend likely reflect the very limited amount of OC-Fe associations at this site, consistent with high OC loadings observed in WGB1 and WGB2, which suggest minimal physical protection. Our findings in the WGB are in line with our previous study in Swedish fjords (Placitu et al., 2024) where we reported low amounts of OC-Fe associations and no consistent trends with sediment depth.

We added this reference in the first paragraph of section 3.3 and the following text: "Similar low and variable amounts of %OC–Fe have recently been reported in Swedish fjord sediments (Placitu et al., 2024)."

Longman, J., Faust, J. C., Bryce, C., Homoky, W. B., & März, C. (2022). Organic Carbon Burial With Reactive Iron Across Global Environments. Global Biogeochemical Cycles, 36(11), e2022GB007447. https://doi.org/10.1029/2022GB007447

Placitu, S., van de Velde, S. J., Hylén, A., Hall, P. O. J., Robertson, E. K., Eriksson, M., Leermakers, M., Mehta, N., and Bonneville, S.: Limited Organic Carbon Burial by the Rusty Carbon Sink in Swedish Fjord Sediments, Journal of Geophysical Research: Biogeosciences, 129, e2024JG008277, https://doi.org/10.1029/2024JG008277, 2024.

Q29: Nothing is mentioned regarding the fact that WGB1 has significantly lower FeR than the other profile. Why? What drives this behavior?

A29: We acknowledge the lower $Fe_R$ content at WGB1 compared to WGB2. This difference is ascribed to a particle shuttling phenomenon that has been described in Nilsson et al., (2021) for the Baltic Sea in which particulate matter is transported from shallow bottoms and deposited in deeper basins. Such redistributions probably explain the lower $Fe_R$ content at the shallower, hypoxic WGB1 compared to the deeper, anoxic WGB2.

Revised text (section 3.1): "In contrast, the absence of a trend with depth in $\delta^{13}$C at WGB2 suggests that OC burial at this station is more strongly influenced by lateral OC inputs. This observation is consistent with the "shuttling" of particulate material described by Nilsson et al., (2021) where repeated cycles of resuspension-redeposition transport particle from shallow, erosive areas (such as WGB1) to deep accumulation areas (such as WGB2)."

Nilsson, M. M., Hylén, A., Ekeroth, N., Kononets, M. Y., Viktorsson, L., Almroth-Rosell, E., Roos, P., Tengberg, A., and Hall, P. O. J.: Particle shuttling and oxidation capacity of sedimentary organic carbon on the Baltic Sea system scale, Marine Chemistry, 232, 103963, https://doi.org/10.1016/j.marchem.2021.103963, 2021.

Q30: How do authors explain the lack of OC-Fe at WGB 2 vs the 5% at WGB1, when the latter has much lower FeR?

A30: We acknowledge the difference in %OC-$Fe_R$ between WGB1 and WGB2. However, our findings are consistent with numerous previous studies showing that neither $Fe_R$ nor SOC concentrations directly control the extent of OC-Fe associations (e.g. Faust et al., 2021; Longman et al., 2021; Sirois et al., 2018; Placitu et al., 2024; Peter and Sobek, 2018). Additional factors such mineralogical variability within the $Fe_R$ pool (i.e. variations in the repartition of Fe mineral phases exhibiting different surface properties toward OC) or differences in the source and quality of OM, as well as the microbial reactivities of OM and Fe phases in OC-Fe associations likely play a more important role in controlling the amount of OC-Fe associations.

Faust, J. C., Tessin, A., Fisher, B. J., Zindorf, M., Papadaki, S., Hendry, K. R., Doyle, K. A., and März, C.: Millennial scale persistence of organic carbon bound to iron in Arctic marine sediments, Nat Commun, 12, 275, https://doi.org/10.1038/s41467-020-20550-0, 2021.

Longman, J., Gernon, T. M., Palmer, M. R., and Manners, H. R.: Tephra Deposition and Bonding With Reactive Oxides Enhances Burial of Organic Carbon in the Bering Sea, Global Biogeochemical Cycles, 35, e2021GB007140, https://doi.org/10.1029/2021GB007140, 2021.

Peter, S. and Sobek, S.: High variability in iron-bound organic carbon among five boreal lake sediments, Biogeochemistry, 139, 19–29, https://doi.org/10.1007/s10533-018-0456-8, 2018.

Placitu, S., van de Velde, S. J., Hylén, A., Hall, P. O. J., Robertson, E. K., Eriksson, M., Leermakers, M., Mehta, N., and Bonneville, S.: Limited Organic Carbon Burial by the Rusty Carbon Sink in Swedish Fjord Sediments, Journal of Geophysical Research: Biogeosciences, 129, e2024JG008277, https://doi.org/10.1029/2024JG008277, 2024.

Sirois, M., Couturier, M., Barber, A., Gélinas, Y., and Chaillou, G.: Interactions between iron and organic carbon in a sandy beach subterranean estuary, Marine Chemistry, 202, 86–96, https://doi.org/10.1016/j.marchem.2018.02.004, 2018.

Q31: If WGB2 is characterized as anoxic, while WGB1 is described as hypoxic, how do the authors explain the higher FeR concentrations at WGB2? This seems counterintuitive, given that more reducing conditions typically promote FeR dissolution.

A31: We agree with the reviewer. As explained in A29, $Fe_R$ rich particles may be transported from shallower, oxic or hypoxic areas (WGB1) to deeper, anoxic portions of the basin (WGB2). Although reducing conditions generally promote the dissolution of $Fe_R$, it also is not uncommon to observe residual $Fe_R$ at depth in Baltic Sea sediment. This results most likely from the formation of Fe(II)-phosphate minerals, such as vivianite-like minerals in WGB2, which is commonly observed in Baltic Sea sediments (e.g. Dijkstra,et al., 2018; Egger et al., 2015; van Helmond et al., 2020; Kubeneck et al., 2021). These Fe(II)-P phases can be extracted by the CBD extraction and probably account for the $Fe_R$ observed in the top sediment in WGB2 (Egger et al., 2015). We have rewritten section 3.3 in relation to WGB 2 to explain this pattern.

Dijkstra, N., Hagens, M., Egger, M., & Slomp, C. P. (2018). Post-depositional formation of vivianite-type minerals alters sediment phosphorus records. Biogeosciences, 15(3), 861–883. https://doi.org/10.5194/bg-15-861-2018

Egger, M., Jilbert, T., Behrends, T., Rivard, C., & Slomp, C. P. (2015). Vivianite is a major sink for phosphorus in methanogenic coastal surface sediments. Geochimica et Cosmochimica Acta, 169, 217–235. https://doi.org/10.1016/j.gca.2015.09.012

Van Helmond, N. A., Slomp, C. P., & Egger, M. (2020). Removal of phosphorus and nitrogen in sediments of the eutrophic Stockholm archipelago, Baltic Sea. Biogeosciences, 17(11), 2745–2762. https://doi.org/10.5194/bg-17-2745-2020

Kubeneck, L., Egger, M., & Slomp, C. P. (2021). Phosphorus burial in vivianite-type minerals in methane-rich coastal sediments. Marine Chemistry, 234, 103948. https://doi.org/10.1016/j.marchem.2021.103948

Revised text (section 3.3) : « At WGB2, despite higher SOC (6.4 ± 4.7%) and $Fe_R$ (average of 40.3 µmol. $g^{-1}$, ranging from 14 to 63 µmol. $g^{-1}$) than in WGB1, only 0.28 %OC-$Fe_R$ (ranging from 0 to 2.8 %OC-$Fe_R$) is found. This is remarkable given that $Fe_R$ concentrations are similar to those reported for other $O_2$-depleted and eutrophic settings such as the Black Sea where $Fe_R$ binds ~15% of the OC (Lalonde et al., 2012). The persistence of $Fe_R$ under anoxic

and sulphidic conditions in WGB2 is intriguing. One potential explanation is the presence of Fe(II)-phosphate mineral phases such as vivianite, which have been well-documented in Baltic Sea sediments and can be solubilized during the CBD extraction (e.g. Dijkstra,et al., 2018; Egger et al., 2015; van Helmond et al., 2020; Kubeneck et al., 2021). "

Q32: How do you reconcile the fact that higher OC concentrations were measured at WGB2, where there is virtually no OC-Fe, and SRR are ~ to those funds at WGB1, but remain constant and not decreasing with depth.

A32: As mentioned in A29 and A30, we ascribe the higher SOC at WGB2 compared to WGB1 to shuttling of particles from shallower to deeper section of the WGB. We disagree that the assessment of the reviewer that the SRR in WGB1 and WGB2 are similar. As a matter of fact, SRR at WGB1 are much higher at 40, 60 and 69 nmol S $cm^{-3} d^{-1}$ vs. 50, 33 and 22 nmol S $cm^{-3} d^{-1}$ at WGB2 for depths 0.5, 1.5 and 2.5 cm. In order to make these difference more obvious, we modified the scale of Figure 2.

Q33: More integration with $\delta^{13}C$ data could clarify early diagenetic pathways but is not attempted.

A33: This is a good suggestion. At WGB1, we observe a progressive $\delta^{13}C$ depletion (and increase of C/N) at depth suggesting a classical, selective microbial degradation targeting labile, N-rich compounds and leaving behind more resistant OM pools. As for WGB2, no systematic depletion trend is observed suggesting that burial is less controlled by selective degradation and more strongly influenced by the "particle shuttling" process (mentioned in A29/A30).

Added text in section 3.1:" At WGB1, progressive $\delta^{13}C$ depletion with depth and increasing C/N indicate selective microbial degradation of labile, N-rich compounds. In contrast, the absence of a trend with depth in $\delta^{13}C$ at WGB2 suggests that OC burial at this station is more strongly influenced by lateral OC inputs. This observation is consistent with the "shuttling" of particulate material described by Nilsson et al., (2021) where repeated cycles of resuspension-redeposition transport particle from shallow, erosive areas (such as WGB1) to deep accumulation areas (such as WGB 2)."

Q34: It is speculated that FeR reduction may be driven by $H_2S$ production, yet it is unclear whether the presented data support this interpretation. If $H_2S$ formation is linked to elevated sulfate reduction rates, one would expect FeR depletion to be most pronounced in the upper sediments where SRR is highest. However, FeR concentrations appear relatively constant in the top 8 cm, with more substantial declines observed only below 10 cm. This pattern raises important questions: Why does FeR persist where SRR is elevated? What mechanisms might explain the delayed reduction of FeR at depth? Furthermore, how do the differing SRR profiles across stations reconcile with the FeR depth trends shown in Figure 5? These discrepancies deserve more thorough discussion to support the proposed mechanisms.

A34: We acknowledge that our data for $Fe_R$ concentrations in WGB2 remain relatively stable in the top 8 cm and decrease below 10 cm despite elevated SRR in the top sediments. As mentioned in A31, we believe that the presence of Fe(II)-phosphate minerals may explain the persistence of a $Fe_R$ pools at depth in WGB2.

Revised text: "The persistence of $Fe_R$ under anoxic and sulphidic conditions in WGB 2 is intriguing. One potential explanation is the presence of Fe(II)-phosphate mineral phases such as vivianite, which have been well-documented in Baltic Sea sediments and can be solubilized during the CBD extraction (e.g. Dijkstra,et al., 2018; Egger et al., 2015; van Helmond et al., 2020; Kubeneck et al., 2021) ".

Q35: Lines 252-254: The reference to the black carbon sink is highly speculative, and it is unrelated to the data presented.

A35: We agree that the link between sulphide promoted Fe reduction, and the formation of OC-Fe-sulphide associations is speculative, we acknowledge this in the manuscript. Our intention was not to present this as a confirmed pathway but to highlight a plausible process to explain the observed downcore decrease in %OC-Fe. We believe this is relevant to the interpretation of our data as previous experimental study have shown that OC-Fe-S association can form under conditions close to those of the Baltic Sea sediment.

Q36: Lines 255-260: Statements about grain-size variation and mineralogical effects are vague. Please be more specific about what these factors are and how they were evaluated in this study.

A36 : We revised the text as follow in section 3.3: "Differences in the %OC-Fe could also be due to mineralogical variations within in the $Fe_R$ pool at depth, as the CBD extraction reduces and solubilizes a wide array of $Fe_R$, such as ferrihydrite, goethite, and lepidocrocite including Fe(II)-phosphates phases, all of which characterized by their small particle size, yet exhibiting large differences in SSA and surface reactivities towards OM moieties (Ghaisas et al., 2021; Egger et al, 2015). »

Q37: Lines 267-268 The phrase "shuttling of particles toward deeper parts of the basins" is unclear. What process is being referred to, and what evidence supports this interpretation?

A37: See A12 and A29 for details on the shuttling of particles towards deeper section of the basin. This concept is now introduced early in section 3.1.

**Additional Comments:**

Q38: Lines 23-24: Clarify what is meant by "large sources of OC." Are you referring to marine organic matter, or another pool?

A38: Bulk sedimentary OC.

Q39: Line 30: Add reference for CO2/O2 regulation

A39: We included a reference to "Berner, R. A. Burial of organic carbon and pyrite sulfur in the modern ocean: Its geochemical and environmental significance. Am. J. Sci. 282, 451–473 (1982)".

Q40: Line 48: Consider specifying "bacterial mineralization" for clarity.

A40: Done, we added "microbial "

Q41: Line 95: Figure 1, invert the colorbar, so that 0 m is at the top and 450 m at the bottom.

A41: Done

Q42: Section 3.2 appears to be missing. The manuscript jumps from Section 3.1 to 3.3.

A42: Thank you, corrected

Q43: Supplementary Fig S1: When working with $\delta^{13}C$ : C:N bi-plots, it is advisable to plot against N/C (not C:N).

A43: Both C/N and N/C can be used, e.g:

Lamb, A., Wilson, G., & Leng, M. (2006). A review of coastal palaeoclimate and relative sea-level reconstructions using [delta] 13C and C/N ratios in organic material. Earth-Science Reviews, 75, 29–57. https://doi.org/10.1016/j.earscirev.2005.10.003

Lien, WY., Chen, CT., Lee, YH. *et al*. Two-stage oxidation of petrogenic organic carbon in a rapidly exhuming small mountainous catchment. *Commun Earth Environ* **6**, 45 (2025). https://doi.org/10.1038/s43247-025-02015-8

Khan, N., Vane, C., Horton, B., Hillier, C., Riding, J., & Kendrick, C. (2015). The application of δ13C, TOC and C/N geochemistry to reconstruct Holocene relative sea levels and paleoenvironments in the Thames Estuary, UK. Journal of Quaternary Science, 30(5), 417–433. https://doi.org/10.1002/jqs.2784

Placitu, S., van de Velde, S. J., Hylén, A., Hall, P. O. J., Robertson, E. K., Eriksson, M., Leermakers, M., Mehta, N., and Bonneville, S.: Limited Organic Carbon Burial by the Rusty Carbon Sink in Swedish Fjord Sediments, Journal of Geophysical Research: Biogeosciences, 129, e2024JG008277, https://doi.org/10.1029/2024JG008277, 2024.

---

## Author Comment (AC3)

**Review 3**

**Review of: Limited physical protection leads to high organic carbon reactivity in anoxic Baltic Sea sediments**

by Placitu et al. 2025

**Recognition**

First of all, I acknowledge the substantial effort undertaken by the authors in conducting this sedimentary biogeochemical study in the Baltic Sea. The collection of sediment cores from five stations across the Western Gotland Basin, combined with the extensive analytical work represents a significant investment. The authors try to tackle a complex and important question regarding the controls on organic carbon preservation in oxygen-depleted marine environments.

My goal in reviewing this manuscript is to improve my understanding of the study, identify potential gaps in the logical framework, and ultimately contribute to enhancing the quality and impact of this research.

**General comment**

This study aims to challenge the traditional paradigm that anoxia inherently promotes OC preservation by examining the role of physical protection mechanisms, specifically mineral surface availability and OC-iron associations, in controlling OC reactivity and burial efficiency. Their approach combines sediment geochemical profiling, sulfate reduction rate (SRR) measurements, and targeted analyses of mineral-organic associations at five stations. The key finding is that despite long-term anoxia, organic carbon remains highly reactive due to limited physical protection, as indicated by high OC loadings and low OC-Fe associations.

A1: While the study's scientific questions are well aligned with the scope of Biogeosciences, the manuscript in its current form tends toward superficial and speculative interpretation. It also lacks sufficient integration of the dataset across all sampled stations. Specifically, the authors do not justify the use of cores collected from other sites than WGB1 and 2. Furthermore, this limited sampling weakens the extrapolation of their results in the current format. A better integration of all results could improve this. The authors do not necessarily need to provide additional data beyond what is available but should explicitly acknowledge these limitations and critically discuss how the restricted spatial coverage could influence the confidence in, and the extrapolation of, their conclusions.

A1: We acknowledge the limitation that OC-Fe extractions were only conducted at WGB1 and WGB2 (see A1 to Reviewer 2). However, these 2 sites were selected because they "bracket" the environmental conditions in the WGB (i.e., relatively shallow vs deep basin, hypoxic vs. anoxic bottom waters). We complemented this dataset with measurements across all 5 stations (OC content, SSA, SRR, $\delta^{13}C$, C/N ratios) that provide complementary constrains on the role of mineral surface in OC burial as evidenced in Figure 4. As mentioned in A1 to Reviewer 2, we have revised the manuscript text to clarify these limitations in the Conclusion section.

Q2: The manuscript is generally clear and well-written but would gain from refining the narrative to center more firmly on the core dataset and better explain the connection with cited literature. The latter would help reduce the impression of superficial and speculative discussion.

A2: We followed the suggestion of the reviewer and substantially modified the section 3 "Results and Discussion" to connect more clearly our dataset with our interpretation and the literature.

With these revisions, the study has strong potential to provide valuable understandings into OC diagenesis in sediments overlayed by anoxic waters.

I therefore recommend major revisions to improve the integration and the discussion of the results, but I consider that this manuscript merits publication in Biogeosciences once these issues are addressed.

Q3: The authors do not necessarily need to provide additional data beyond what is available but should explicitly acknowledge these limitations and critically discuss how the restricted spatial coverage could influence the confidence in, and the extrapolation of, their conclusions.

A3: We agree that it is important to acknowledge the limitations of our study, particular regarding the spatial coverage. We have made clear in several instances in the manuscript (Introduction, Section 2.3, and in the Conclusions) that the OC-Fe dataset is restricted to two stations (WGB1 and WGB2). As mentioned in A1 to reviewer 2, we deliberately choose these sites because they capture the range of redox and depositional conditions present in WGB. This makes them broadly representative of the main processes controlling OC burial in the basin. In addition, we also emphasize that OC-Fe associations are only one pathway of OC physical protection. To address this process more comprehensively, we complemented our OC-Fe dataset with the quantification of OC loading (i.e., OC/SSA ratios) across all stations which provide a more integrative proxy of the physical protection potential of these sediments. Both OC-Fe and OC/SSA datasets converge to show the limited extent of physical protection in the WGB. Nevertheless, we agree that further work including additional sites would be highly valuable to strengthen our findings.

**Specific comment**

Q4: L82: Please consider using "sedimentary organic carbon (SOC)" rather than POC or sedimentary POC

A4: We agree with the proposed change and have replaced "particulate organic carbon (POC)" with "sedimentary organic carbon (SOC)" throughout the manuscript.

Q5: Could you please provide more details about the sampling of the two subcores? Specifically, how and where were they collected?

A5: Additional details in the text were included in the text " Two subcores (2.5 cm inner diameter, 20 cm length) were collected from the GEMAX corer at each station for sulphate reduction rate (SRR) measurements using the $^{35}$S radiotracer method (Jørgensen, 1978) immediately on retrieval, and the tracer addition and incubation started within 15 minutes of core collection. Duplicate cores were collected from two different GEMAX casts."

Q6: There are many acronyms. I think that, for better readability, it would be preferable to redefine them in each section (Introduction, Methods, Results)

A6: We redefined the acronyms in the different sections as suggested.

Q7: Figure1: Depth should be positive

A7: Corrected

Q8: Table1: Is WGB1 constantly hypoxic or does it oscillate between hypoxic and anoxic conditions?

A8: We did not find conclusive evidence in the literature that WGB1 is affected regularly by anoxia. Long term monitoring data (15 years mean) from the Swedish Meteorological and Hydrological Institute (SMHI) indicate that the hypoxic level ($O_2 < 2$ mg L$^{-1}$) in the WGB is generally situated around 75-80 m depth which corresponds to

the depth of WGB1 (75 m). This suggest that this station is typically subjected to hypoxic conditions. On the other hand, inflows of oxygenated waters (major Baltic inflows) do not always reach the WGB.

Q9: Do you have data from another site (WGB3 to 5) to check for potential environmental gradient?

A9: As mentioned in A3, we have only collected cores for the OC-Fe investigations at WGB1 and WGB2. However, we agree that additional sampling and cores would be highly valuable to strengthen our findings.

Q10: L146: Why only sulfate reduction and no other anoxic process using nitrate or Fe and Mn-oxydes

A10: In persistently anoxic sediments, the contribution of denitrification, DNRA, manganese and iron reduction to organic carbon mineralization is negligible compared to sulphate reduction, given the dominance of sulphate in the porewater geochemistry. If these pathways were quantitatively important, the total reactivity would be even higher than our estimates, meaning our current mineralization rates are likely underestimated.

We included an additional sentence in the main text "These sediments are anoxic, unaffected by bioturbation, and sulphate $SO_4^{2-}$ is not depleted, so one can assume that all OC mineralization results from sulphate reduction. Contributions from denitrification, DNRA, manganese and iron reduction are negligible; however, it is important to note that the reported rates may slightly underestimate total reactivity"

Q11: L149-150: These sediments are not all anoxic (WGB1). Did any trace of biological activity was visible?

A11: We did not investigate *macro*-biological (life higher than that of microbes) activity at any station. Because of the hypoxic conditions at WGB1, we expect that any effect of bioturbation is minimal.

Q12: L164-165: Could you please explain your reasoning behind your interpretation of « fresh material »?

A12: The word fresh was modified '' autochthonous freshwater and marine material''

Q13: L176: Please add the unit

A13: Done

Q14: L174-185: From my understanding, you didn't use the same formula as Katsev and Crowe (2015). Could that affect the location of your data point relative to the two regression lines (oxic/anoxic) derived from Katsev and Crowe (2015)?

A14: Please see A6 in the answer to Reviewer 1.

Q15: Figure3 (and 4): "individual points represent different sediment depths…" Could you find a way to depict depth in this figure, perhaps using a colour scale?

A15: We have now modified Figure 3 and 4 to include a colour code for depth.

Q16: L205: remove the ("(albeit all …"

A16: It is correct, we included the closing '')''.

Q17: L207-208: While I agree with that, the water column is anoxic in your study. I'm curious to know how resuspension events could influence OC mineralization process in anoxic water.

A17: Under typical conditions, shear stress at the bottom of deep anoxic basins of the Baltic Sea, and resuspension is unlikely. However, during episodic events such as major Baltic inflows, resuspension could redistribute sedimentary organic carbon into more microbially active layers, potentially enhancing microbial degradation. Such

events may also expose organic compounds to reactive mineral surfaces or disrupt sediment aggregates, affecting OC–mineral associations. A detailed assessment of these processes is beyond the scope of this study.

Q18: Are all the data points showed? There are fewer than in Figure 3. Why is that?

A18: Figure 3 includes all available data across stations ad depths, whereas Figure 4 is restricted to samples where both SOC and SSA were measured, which explains the smaller dataset.

Q19: L229-231: Do you have an explanation for this high OC loading? eutrophication of the Baltic Sea, large extent of the anoxic water mass, …?

A19: The high OC loadings are due to the high SOC concentrations observed in the Baltic Sea stemming from the occurrence of cyanobacteria, diatoms, and other phytoplankton blooms induced by high nutrients inputs (reflected in the $\delta^{13}$C signature and C/N ratios of WGB1 and WGB2 sediments). Due to increased nutrient loading and eutrophication since the mid 1900's, the OC input has increased considerably in the central Baltic Sea, and at the same time, the construction of dams in the main Baltic rivers has caused a decrease of the suspended particle input. See A13 in answer to Reviewer 1 for complementary details on this matter.

Q20: L240-241: Why not the OC reactivity rather than OC loading, which could allow for intense mineralization? Or maybe, does the increase of OC loading at the sediment surface results from relatively "recent" eutrophication rather than OC mineralization?

A20: Please see A13 in the answer to Reviewer 1.

---

## Referee Report (RR1)

**Second Revision: Limited physical protection leads to high organic carbon reactivity in anoxic Baltic Sea sediments**

I sincerely appreciate substantial revision from the authors. The manuscript is now significantly improved and almost ready for publication. The key scientific discussion is now solid. I only have few comments as listed below.

**Line 42-43:** The phrase "active preservation mechanism of anoxia" is not clear. What's about replacing this phrase with "effective preservation of organic carbon in the anoxic environments".

**Line 184-186:** To illustrate these points, I'd suggest the authors to show plots of downcore changes in $\delta^{13}C$ and C/N in supplementary materials. The Supplementary Fig. 1 which has already portrayed $\delta^{13}C$ and C/N data does not contain depth information.

While I agree with the authors that irregular changes in $\delta^{13}C$ throughout the core WGB2 possibly resulted from lateral inputs of OC, another possible cause of $\delta^{13}C$ fluctuation is the changing rate of microbial decomposition of OC, as evidenced by fluctuating SRR across the core WGB2 (please see my comments in line 195-196).

**Line 190:** "SOC profiles heterogeneity" or "heterogeneity of SOC profiles"

**Line 195-196, Figure 2g:** Do you think that the absence of SRR-depth trend in core WGB2 is correlated with the absence of $\delta^{13}C$-depth trend in core WGB2 (line 185-186)? Fluctuation in $\delta^{13}C$, which was caused by microbial transformation of original $\delta^{13}C$ signal, probably suggests that microbial activities is not uniform throughout the entire core.

As the authors mentioned earlier that core WGB2 is located in the depocenter which received materials from multiple resuspension-redeposition (Nilsson et al., 2021)). Each sediment layer in core WGB2 may receive "already degraded" materials from other locations. Some sediment layers may not contain easily degradable OC anymore while other sediment layers may receive fresher materials. This may lead to the fluctuation of SRR across the entire core.

**Line 209-210:** " stations with higher SRR (e.g., WGB3) plotting nearer to the oxic relationship, while those with lower SRR (e.g., WGB2) tend toward the anoxic trend"

Personally, I do not see these trends when looking at Figure 3. I'd suggest the authors to provide some numerical parameters (e.g., average vertical distance from data points to the anoxic and oxic trendline) to strengthen their statement.

**Line 269:** "strongly contrast with..."

**Line 276:** Zhao et al. (2018) discovered that frequent physical reworking (in mobile mud belt) can destroy the association between FeR and OC. This concept may explain the very low amount of OC-$Fe_R$ in core WGB2 which underwent multiple cycles of sediment redistribution, as evidenced by highly fluctuated $\delta^{13}C$ and SRR as shown in previous paragraphs and past literature (Nilsson et al., 2021).

Zhao, B., Yao, P., Bianchi, T. S., Shields, M. R., Cui, X., Zhang, X., ... & Yu, Z. (2018). The role of reactive iron in the preservation of terrestrial organic carbon in estuarine sediments. Journal of Geophysical Research: Biogeosciences, 123(12), 3556-3569.

**Line 304:** "...supporting the hypothesis that $Fe_R$ preferentially binds terrestrial organic matter..."

Since the current study did not investigate chemical composition of $OC$-$Fe_R$ to indicate their source, I think it is overinterpreted to conclude that "$Fe_R$ preferentially binds terrestrial organic matter". The current study suggested that $Fe_R$ that had been pre-formed on land could not incorporate OC that was formed later in the Baltic Sea into OC-FeR association.

Therefore, I'd suggest removing this phrase.

---

## Author Response (AR2)

**Second Revision: Limited physical protection leads to high organic carbon reactivity in anoxic Baltic Sea sediments**

I sincerely appreciate substantial revision from the authors. The manuscript is now significantly improved and almost ready for publication. The key scientific discussion is now solid. I only have few comments as listed below.

Thank you for your time and efforts!

**Q1 Line 42-43**: The phrase "active preservation mechanism of anoxia" is not clear. What's about replacing this phrase with "effective preservation of organic carbon in the anoxic environments".

A1: We modified this sentence:

Revised text: "This pattern is generally attributed not to the effective preservation of organic carbon in anoxic environments, but rather as the result of $O_2$ strongly enhancing OC mineralization when present."

**Q2 Line 184-186**: To illustrate these points, I'd suggest the authors to show plots of downcore changes in δ13C and C/N in supplementary materials. The Supplementary Fig. 1 which has already portrayed δ13C and C/N data does not contain depth information. While I agree with the authors that irregular changes in δ13C throughout the core WGB2 possibly resulted from lateral inputs of OC, another possible cause of δ13C fluctuation is the changing rate of microbial decomposition of OC, as evidenced by fluctuating SRR across the core WGB2 (please see my comments in line 195-196).

A2: We thank the reviewer for the suggestion. We examined the SRR and $\delta^{13}C$ records and found no clear covariation in WGB 2 (see updated Supplementary Fig. S1). This supports our interpretation that lateral inputs, rather than downcore changes in microbial degradation rate, are the dominant control on $\delta^{13}C$ variability in WGB2. To clarify this point, we added a few sentences in section 3.1.

Revised text: "At WGB2, $\delta^{13}C$ values fluctuate irregularly with depth (Figure S1 in Supplementary information). Such variability could, in principle, reflect changing rates of microbial SOC degradation, as isotopic fractionation during microbial metabolism generally enriches the residual organic matter in $^{13}C$. However, the $\delta^{13}C$ and SRR profiles in WGB2 show no consistent covariation, indicating that variations in microbial activity are unlikely to be the primary driver of the $\delta^{13}C$ pattern. Instead, the variations of $\delta^{13}C$, C/N and SRR most likely reflect heterogeneity in the deposited material: certain layers may contain more reactive OC that transiently enhances microbial activity, whereas others consist of more degraded OC that experienced substantial mineralization during lateral transport. This interpretation aligns with the WGB sedimentation regime, where lateral transport involves repeated resuspension from shallow, erosive regions (e.g., WGB1) and redeposition in deeper, accumulating depocenters (e.g., WGB2; Nilsson et al., 2021)."

**Q3 Line 190**: "SOC profiles heterogeneity" or "heterogeneity of SOC profiles"

A3 : We changed this sentence.

**Q4 Line 195-196**, Figure 2g: Do you think that the absence of SRR-depth trend in core WGB2 is correlated with the absence of δ13C-depth trend in core WGB2 (line 185-186)?

A4 : We explored this point in A2.

**Q5** : Fluctuation in δ13C, which was caused by microbial transformation of original δ13C signal, probably suggests that microbial activities is not uniform throughout the entire core. As the authors mentioned earlier that core WGB2 is located in the depocenter which received materials from multiple resuspension-redeposition (Nilsson et al., 2021)). Each sediment layer in core WGB2 may receive "already degraded" materials from other locations. Some sediment layers may not contain easily degradable OC anymore while other sediment layers may receive fresher materials. This may lead to the fluctuation of SRR across the entire core.

A5: This is a good point. We agree that fluctuations in $\delta^{13}C$, C/N at WGB2 may in part reflect spatial and temporal heterogeneity in OC deposited at this site, with some sediment layers receiving more degraded material while others contain more reactive OC that transiently supports microbial activity. We have now revised the section 3.1 to explicitly acknowledge this possibility.

Revised text: "At WGB2, $\delta^{13}C$ values fluctuate irregularly with depth (Figure S1). Such variability could, in principle, reflect changing rates of microbial SOC degradation, as isotopic fractionation during microbial metabolism generally enriches the residual organic matter in $^{13}C$. However, the $\delta^{13}C$ and SRR profiles in WGB2 show no consistent co-variation, indicating that variations in microbial activity are unlikely to be the primary driver of the $\delta^{13}C$ pattern. Instead, the variations of $\delta^{13}C$, C/N and SRR most likely reflect heterogeneity in the deposited material: certain layers may contain more reactive OC that transiently enhances microbial activity, whereas others consist of more degraded OC that experienced substantial mineralization during lateral transport. This interpretation aligns with the WGB sedimentation regime, where lateral transport involves repeated resuspension from shallow, erosive regions (e.g., WGB1) and redeposition in deeper, accumulating depocenters (e.g., WGB2; Nilsson et al., 2021)"

**Q6 Line 209-210**: " stations with higher SRR (e.g., WGB3) plotting nearer to the oxic relationship, while those with lower SRR (e.g., WGB2) tend toward the anoxic trend" Personally, I do not see these trends when looking at Figure 3. I'd suggest the authors to provide some numerical parameters (e.g., average vertical distance from data points to the anoxic and oxic trendline) to strengthen their statement.

A6: We thank the reviewer for this thoughtful comment. We have quantitatively evaluated the proximity of data points from each station to the oxic and anoxic trendlines in Figure 3. We calculated the Root Mean Squared Error (RMSE) from each data point to both the oxic and anoxic trendlines, providing a robust numerical measure of how closely our dataset aligns with the oxic trendline.

Revised text: "Root Mean Squared Error (RMSE) for the anoxic trendline and the k vs. 210Pb-estimated OC age dataset is 0.68, while the RMSE for the oxic trendline was significantly lower at 0.45. This indicates that, overall, the sediment reactivity more closely aligns with the oxic trendline."

**Q7 Line 269**: "strongly contrast with…"

A7 : Done.

**Q8 Line 276**: Zhao et al. (2018) discovered that frequent physical reworking (in mobile mud belt) can destroy the association between FeR and OC. This concept may explain the very low amount of OC-FeR in core WGB2 which underwent multiple cycles of sediment redistribution, as evidenced by highly fluctuated δ13C and SRR as shown in previous paragraphs and past literature (Nilsson et al., 2021).

Zhao, B., Yao, P., Bianchi, T. S., Shields, M. R., Cui, X., Zhang, X., … & Yu, Z. (2018). The role of reactive iron in the preservation of terrestrial organic carbon in estuarine sediments. Journal of

Geophysical Research: Biogeosciences, 123(12), 3556-3569.

**A8** : We thank the reviewer for this valuable suggestion. We agree that frequent physical reworking and sediment redistribution could contribute to the very low proportion of OC-Fe observed in WGB2. Repeated resuspension–redeposition events and fluctuating redox conditions may disrupt OC-Fe associations and enhance their decoupling, as reported for mobile mud systems by Zhao et al. (2018). We have now incorporated this interpretation into the discussion (section 3.3 – lines 304-308).
Revised text: "The very low %OC-Fe at WGB2 may reflect the effects of repeated physical reworking and sediment redistribution prior to deposition in WGB2. Frequent resuspension and redeposition events can disrupt the OC-Fe associations by exposing particles to fluctuating redox conditions and mechanical disaggregation, as observed in mobile mud environments (Zhao et al., 2018). This process is consistent with the strong downcore variability in $\delta^{13}C$ and SRR at WGB2 and with earlier evidence of sediment shuttling and lateral transport in the area (Nilsson et al., 2021)."

**Q9 Line 304**: "…supporting the hypothesis that FeR preferentially binds terrestrial organic matter…"
Since the current study did not investigate chemical composition of OC-FeR to indicate their source, I think it is overinterpreted to conclude that "FeR preferrentially binds terrestrial organic matter". The current study suggested that FeR that had been pre-formed on land could not incorporate OC that was formed later in the Baltic Sea into OC-FeR association. Therefore, I'd suggest removing this phrase.

A9: We agree that our data do not allow direct inference about the source or composition of the OC moieties in OC–Fe associations and that the statement suggesting that "$Fe_R$ preferentially binds terrestrial organic matter" was too speculative, although it is supporting by a range of studies. We have removed this sentence, and the paragraph focus instead on the decoupled delivery of $Fe_R$ and OC and the predominance of autochthonous OC production in the WGB which together likely limit the formation of OC–Fe associations.